# Injectables and Depots to Prolong Drug Action of Proteins and Peptides

**DOI:** 10.3390/pharmaceutics12100999

**Published:** 2020-10-21

**Authors:** Nkiruka Ibeanu, Raphael Egbu, Lesley Onyekuru, Hoda Javaheri, Peng Tee Khaw, Gareth R. Williams, Steve Brocchini, Sahar Awwad

**Affiliations:** 1School of Pharmacy, University College London, 29-39 Brunswick Square, London WC1N 1AX, UK; nkiruka.ibeanu.17@ucl.ac.uk (N.I.); raphael.egbu.16@ucl.ac.uk (R.E.); lesley.onyekuru.11@ucl.ac.uk (L.O.); h.javaheri@ucl.ac.uk (H.J.); g.williams@ucl.ac.uk (G.R.W.); s.brocchini@ucl.ac.uk (S.B.); 2National Institute for Health Research (NIHR) Biomedical Research Centre at Moorfields Eye Hospital NHS Foundation Trust and UCL Institute of Ophthalmology, London EC1V 9EL, UK; p.khaw@ucl.ac.uk

**Keywords:** half-life, drug delivery, ocular, subcutaneous, intravenous, oral, mucosal, biologics, proteins, peptides

## Abstract

Proteins and peptides have emerged in recent years to treat a wide range of multifaceted diseases such as cancer, diabetes and inflammation. The emergence of polypeptides has yielded advancements in the fields of biopharmaceutical production and formulation. Polypeptides often display poor pharmacokinetics, limited permeability across biological barriers, suboptimal biodistribution, and some proclivity for immunogenicity. Frequent administration of polypeptides is generally required to maintain adequate therapeutic levels, which can limit efficacy and compliance while increasing adverse reactions. Many strategies to increase the duration of action of therapeutic polypeptides have been described with many clinical products having been developed. This review describes approaches to optimise polypeptide delivery organised by the commonly used routes of administration. Future innovations in formulation may hold the key to the continued successful development of proteins and peptides with optimal clinical properties.

## 1. Introduction

Polypeptides (i.e., proteins and peptides) play fundamental roles in most biological processes, and their therapeutic use has revolutionised the treatment of many diseases, e.g., diabetes, hepatitis, rheumatoid arthritis, cancer, infection, inflammation and multiple sclerosis. Significant numbers of these molecules have been clinical approved and many more are in clinical development (e.g., monoclonal antibodies (mAbs), antibody-drug conjugates (ADCs), interferons (IFNs), interleukins (ILs), enzymes, hormones, blood factors, engineered scaffolds, vaccines and protein fusions) [1,2,3,4,5]. Polypeptide make up about 10% of marketed drugs [6,7]. In terms of market share, the global peptides market was valued at USD 21.5 billion as of 2016 and expected to grow at a compound annual growth rate (CAGR) of 9.4% up to the year 2025 [8]. The global therapeutic proteins market was valued at about USD 93.1 billion as of 2018 with an expected grow rate of 16.7% through 2022 [9].

Proteins are highly potent molecules compared to most low molecular weight molecules [10]. Drug development and isolation from natural sources (e.g., insulin and adrenocorticotrophic hormone (ACTH)) or via chemical synthesis (e.g., synthetic oxytocin and vasopressin) have enabled the rapid identification and development of numerous clinical peptides [11]. Proteins and peptides are distinguished from each other by differences in molecular volume and the complexity of non-covalent interactions. Peptides range from 2 to 100 amino acids, while proteins generally comprise more than 100 amino acids [12]. Peptides typically must preserve their secondary structure while proteins must retain their tertiary structure, and sometimes quaternary structure to maintain biological activity.

Excluding antibody-based medicines, most other therapeutic polypeptides act to replace a corresponding endogenous polypeptide (e.g., insulin, blood factors and interferons (IFNs)). As with low molecular weight drugs, improved versions of clinically used polypeptides are developed, often as a means for lifecycle management. Dosage forms and/or pharmacokinetic profiles are often improved with new product versions. Pharmacokinetic improvements often include modification of the amino acid sequence in the polypeptide, so the biobetter versions of a replacement polypeptide will be compositionally different than the endogenous polypeptide.

Non-endogenous polypeptides are also used clinically and are being widely evaluated in preclinical programs. Many different non-endogenous polypeptides have been described [13] such as designed ankyrin repeat proteins (DARPins), single-domain antibodies (e.g., nanobodies (V_H_H antibodies), domain antibodies, anticalins, avimers, adnectins and bispecifics such as bispecific T-cell engagers (BiTEs), tandem diabodies (TandAbs) and dual-affinity re-targeting antibodies (DARTs) [14] (Figure 1). Since many of these scaffolds have short serum half-lives, strategies to improve their pharmacokinetic properties are necessary [14].

Therapeutic polypeptides often suffer from loss of their biological activity due to misfolding and aggregation, lack of stability to hydrolysis and enzymatic degradation, and suboptimal pharmacokinetics. Dose dumping, low therapeutic levels, increased adverse reactions, increased costs and poor patient compliance characterise the consequences for suboptimal pharmacokinetic properties (Section 2) [15]. A more optimised pharmacokinetic profile of polypeptides is often the basis for lifecycle management.

Different strategies (Section 3) are described to prolong the duration of action and circulation half-lives of polypeptides. Unlike therapeutic proteins, most peptides lack significant tertiary structure, which allows them to be formulated as depots (e.g., Bydureon^®^) provided that secondary structure can be maintained, and fibrillation avoided. Other strategies can be used for both peptides and proteins, e.g., albumin binding and possibly fragment crystallisable (Fc) fusions. Routes of administration also play a crucial role that influences strategies to optimise pharmacokinetic profiles and the duration of drug action. The purpose of this review is to describe general challenges and strategies to develop long-acting polypeptide formulations. We then focus on the influence that the key routes of administration have for developing long-acting dosage forms of polypeptides.

## 2. General Challenges in Polypeptide Delivery

### 2.1. Aggregation

Polypeptide aggregation is one of the most challenging issues encountered in almost all phases of manufacture and development [16]. Disruption of protein tertiary structure can increase protein–protein interactions to cause thermodynamically-driven aggregation. Loss of tertiary structure can also result in protein misfolding and denaturation [17,18]. Peptide self-assembly is a concept that has found applications in the treatment of various diseases [19] based on the ability of peptides to self-associate in response to environmental conditions such as pH, peptide concentrations and amino acid sequence [16]. Lanreotide, for example, is a synthetic analogue of somatostatin for the treatment of acromegaly, which self-assembles in water into monodisperse nanotubes [20], available in a controlled release delivery system. However, as is the case with numerous other peptides, their propensity to self-associate may lead to the formation of fibrils and aggregates with consequently reduced activity and bioavailability [21]. These aggregates can be amorphous or highly structured forms (e.g., amyloid fibrils, which self-associate to form oligomeric structures and a critical nucleus) [16]. Amyloid fibril formation can result in irreversible aggregates, and external factors (e.g., gentle heat or pH) cannot revert this change [22].

Factors such as changes in ionic strength, pH, temperature, light, pressure and mechanical stress can result in an increased exposure of hydrophobic residues, which are often not solvent accessible, and which can drive protein aggregation. Extreme conditions of pH and temperature can result in chemical degradation including beta-elimination, racemisation and hydrolysis, which can also cause aggregate to occur [23]. As pH affects the solubility of proteins, at the isoelectric point a protein will generally display reduced solubility, which can also drive aggregation formation [24]. Another factor that contributes to aggregation is polypeptide concentration [16,25] in solution and at interfaces at surfaces or with materials [26,27]. Manufacturing processes (e.g., fermentation, purification, formulation, filling, shipment and storage) of biopharmaceuticals can also lead to aggregation, especially upstream prior to formulation [28]. Aggregation can occur due to shear and filling steps, multiple freeze–thaw cycles, compounding and mechanical shock [29,30]. Aggregation also presents significant challenges in manufacturing due to reduction in yields [31].

Aggregation poses a risk for clinical use due to reduced dose reproducibility and stimulation of an undesired immune response [32,33]. Aggregation within the dosage form usually manifests as a loss of activity [34] or binding affinity [35]. There are also concerns that irreversible protein aggregates can result in patients becoming immune to native folded protein or developing an autoimmune disease [36]. A clinical investigation by Moore and Leppert [37] studied the presence and development of antibodies in patients after the administration of human growth hormone (hGH) with different aggregation levels. Antibody development was reported to be dependent on the presence of hGH aggregates, and the antibody response was likely to be transient with <10% aggregation [37].

Severe reactions can occur with elevated immunogenic responses, for example, anaphylactic shock related to the use of a chimeric anti-IL2 receptor antibody has been reported [38]. The development of immunoglobulin E (IgE) was noted to trigger anaphylaxis upon second exposure to the drug, which was further exacerbated by aggregation [38]. Peptide aggregates have also been associated with a number of disease states; for instance, insulin aggregation, and the subsequent formation of amyloid fibrils, is known to result in insulin injection amyloidosis and injection site immune reactions as well as poor glycaemic control in many cases [39,40]. Fibrillation of Alzheimer’s related peptide is also associated with the progression of the disease [41].

### 2.2. Pharmacokinetics

Most polypeptides have suboptimal in vivo half-lives for effective therapy resulting in frequent dosing. Each route of administration (Section 4) has different factors that contribute to short in vivo half-lives; for example, the fluid turnover in the eye affects the half-life of intraocularly administered drugs; whereas degradation (by enzymes or pH) is more prevalent by the oral and subcutaneous (SC) routes. Oral absorption of polypeptides is additionally hampered by the molecular weight and charge of proteins, which significantly reduces permeation to limit tissue distribution.

Once in the blood compartment after parenteral administration, the solution size of a polypeptide also influences the renal clearance [42]. The kidney glomeruli have a pore size of about 8 nm [43] so they allow the passage of peptides and many therapeutic proteins, which are rapidly filtered through the kidney. Since these polypeptides are not reabsorbed in the renal tubules, they tend to have short half-lives [44].

Enzymatic degradation can also reduce the pharmacokinetics and bioavailability of biopharmaceuticals at the site of administration or target location [45,46]. The slow process of lymphatic transport following SC administration, for instance, makes the administered polypeptide more susceptible to degradation by enzymes [45]. Biopharmaceuticals can also be affected by pre-systemic enzyme degradation by proteases and hydrolases [47].

Polypeptides are zwitterions, so charge is a complex and a heterogenous physicochemical property, which may affect the pharmacokinetic profile. Interactions between biotherapeutics structural macromolecules such as collagen, elastin, glycoproteins and microfibrillar proteins in the extracellular matrix (ECM) of the hypodermis may delay absorption following SC administration [48]. Peptides could also undergo non-specific binding to endogenous proteins, which also affects their absorption; for example, liraglutide shows more than 98% binding to proteins in the plasma, while octreotide is known to bind to lipoproteins (~65% binding) [46].

Administration of proteins (such as human mAbs, non-human proteins and blood factors) can lead to the development of anti-drug antibodies (ADAs). The immunogenic potential of a chimeric antibody is more likely to induce immunogenicity than a fully humanised immunoglobulin [49]. Immune complexes form when protein therapeutics bind to ADAs leading to protein elimination and the loss of being able to administer a reproducible dose [50]. Many factors contribute to the formation of ADAs including disease and patient-related considerations, the presence of aggregates, duration of treatment, the dose and route of administration. There may also be product-related factors implicated in protein immunogenicity, such as post-translational modification, changes due to storage conditions, protein degradation, and conformational changes [50]. ADAs could either be neutralising or non-neutralising to the protein-implying loss or retention of protein function, respectively, which can alter the pharmacokinetic profile or sometimes lead to an increased elimination [51], ultimately affecting the efficacy and toxicity profile of the biologic [50,52,53].

## 3. General Strategies to Increase Duration of Action

### 3.1. Half-Life Extension Strategies

Rapid elimination in the blood compartment remains one of the major concerns in the development and lifecycle management of therapeutic polypeptides. Technologies for half-life extension are based on genetic engineering and post-translational modification of the biomolecule (e.g., bioconjugation and glycosylation) [54]. Increasingly it is becoming apparent that protein engineering and formulation can work hand in hand to increase the duration of drug action [55]. Increased stability and circulation times can sometimes be achieved, especially for peptides to degradation to peptidases by amino acid substitution to remove endopeptidase sites (e.g., exenatide) or to utilise D-amino acids (e.g., octreotide).

#### 3.1.1. Recycling

One function of the Fc region of an antibody is to recycle the antibody while it circulates in the blood. The neonatal Fc receptor (FcRn) comprises a heavy (α) chain, the glycosylated FcRn, and beta-2-microglobulin (β2M) light chain. The FcRn plays a crucial role in mediating transplacental immunoglobulin transport from a mother’s milk to breast-fed infants. The FcRn also mediates the recycling and transport of albumin and immunoglobulin throughout life. Antibody Fc binds to endothelial cell FcRn during endocytosis with high affinity in the endosome at an acidic pH (~6.5). As the endosome recycles to the plasma membrane, there is a decreased affinity of antibody Fc to the FcRn at physiological pH. The antibody can then dissociate from FcRn back into the bloodstream. Fc recycling results in prolonging the therapeutic antibody circulation half-life in a dose-dependent manner.

Fc-fusion polypeptides are made by recombinantly fusing two or more polypeptides (e.g., an extracellular receptor domain) to an Fc to exploit this recycling pathway [56]. One challenge is the difficulty in developing suitable stable linker regions from the Fc region to the fused polypeptide to allow efficient interaction to the target ligand or receptor [57]. Enbrel^®^ (etanercept), which targets TNF-α, is used to treat rheumatoid arthritis was the first blockbuster Fc-fusion product). Other Fc-fusion proteins include abatacept (Orencia^®^), aflibercept (Eylea^®^), alefacept (Amevive^®^), belatacept (Nulojix^®^), rilonacept (Arcalyst^®^) and ziv-aflibercept (Zaltrap^®^) [58]. Peptibodies are Fc fusions of peptides and approved products include romiplostim (Nplate^®^) and dulaglutide (Trulicity^®^). Eftrenonacog-α (Alprolix^®^) and efraloctocog-α (Eloctate^®^) are Fc-fusion products of blood factors to treat haemophilia [58].

#### 3.1.2. Increasing Size and Shielding Effects

##### Glycoengineering

Post-translational glycosylation or glycoengineering can extend the half-life of biotherapeutics [59,60]. Glycosylation may also improve the molecular stability of peptides or modify the conformation of the peptide backbone [61]. Glycoengineering involves the covalent binding of a carbohydrate chain to the polypeptide surface as a post-translational modification during protein synthesis. It is reported to increase the in vivo molecular stability of proteins once administered by prevention of proteolytic degradation [62]. Rapid clearance is decreased by the presence of galactose-terminating glycans through asialoglycoprotein-possessing glycans terminating in mannose, N-acetylglucosamine or fucose through leucine-like receptors. The most prevalent glycosylation sites occur at asparagine (N-linked) and serine/threonine (O-linked) residues. The surface charge and isoelectric point of a protein can be altered by functionalising the end of glycan core structures with phosphate, sulphate and carboxylic acid with chemically charged glycans like sialic acid [62,63]. As a result, the half-life of glycoproteins can be extended. Chemical and chemo-enzymatic glycosylation methods have also been described and pursued as alternatives to N- and O-linked glycosylation [61].

A prominent example of the use of this concept is hyperglycosylated erythropoietin (EPO), marketed as Aranesp^®^ (darbepoetin-α) and used for the treatment of anaemia associated with myelosuppressive chemotherapy and chronic renal insufficiency [64]. Darbepoetin-α was engineered to contain two additional glycosylation sites (first-generation erythropoietin (EPO) possesses three N-linked glycosylation sites) to increase the size of EPO. Darbepoetin-α has proven itself to be clinically beneficial. Compared to the unmodified protein, darbepoetin-α exhibits a 3-fold extended half-life, permitting dosing once weekly or once every two weeks [65]. A synthetic erythropoiesis protein (SEP) has been described by Kochendoefer et al. [66] as an alternative to other glycosylated EPO motifs produced recombinantly [66]. Compared to recombinant EPO, SEP had a 2-fold increased half-life (9.5 h) following IV administration in rats [66,67]. The modification of IFN-β1a by N-glycosylation also led to a moderate increase in half-life in comparison to native IFN [68].

Glycosylation is associated with a few challenges due to difficulties in using expression systems for glycoprotein production. Structural heterogeneity and low yields are typical of host-expression systems, and a further understanding of the mechanisms by which glycosylation affects the in vivo properties of peptides is still required for rational drug design [61]. However, this technology provides numerous future opportunities towards improving the therapeutic efficacy of polypeptides [62].

##### Polypeptide Conjugation to a Water-Soluble Macromolecule

The covalent conjugation of the water-soluble polymer poly(ethylene glycol) (PEG) to a therapeutic polypeptide can markedly increase circulation time by a range of mechanisms including (i) decreasing renal clearance rates due to the increased size of the conjugate and (ii) inhibiting enzymatic-mediated degradation and endocytic clearance due to PEG steric shielding. There is often a reduction in protein biological efficacy due to PEG steric shielding, which can be mitigated by site selective conjugation [55]. PEGylation is a clinically proven approach to improve the efficacy of many therapeutic proteins and enzymes, and several PEGylated products (including biosimilars) have become first-line treatments [69] with many PEGylated products continuing to be developed. The detail of protein PEGylation has been described in many recent reviews and books [70,71].

Some concerns regarding the development of secondary antibodies to PEG [72,73] do exist especially for non-human proteins that are conjugated to many molecules of PEG [74,75,76,77,78]. The majority of products are mono-PEGylated products so the only reported toxicities have been associated with the protein [79]. There have also been reports about the generation of kidney vacuoles with the use of high doses of PEG in animals, and it has been seen that this issue ceases when the dosing of PEG is stopped [10,80,81]. However, kidney vacuoles associated with PEG have not been reported in humans [82,83]. Administered doses of PEGylated protein conjugates are higher in animals compared to humans [80,84], as doses in humans tend to be low, e.g., 180 μg for PEGasys^®^ (except for certolizumab pegol at a 200 mg dose); therefore, chances of PEG accumulation and toxicity are low [71].

Alternative macromolecules to PEG have been described, including poly(sialic acid) (polysialylation), poly(glutamic acid) (glutamylation), homo-amino acid polymer (HAPylation), heparosan polymer (HEPylation), hydroxyethyl starch (HESylation), proline-alanine-serine repeats (PASylation) and unstructured polypeptides (XTENylation). Several of these strategies listed involve appending the half-life extending macromolecule during polypeptide expression, which avoids chemical conjugation during downstream processing (e.g., XTEN and PASylation) [85,86,87].

Human serum albumin (HSA) is the most prevalent protein in blood and has a serum half-life of 19–22 days due to recycling by FcRn interactions similar to immunoglobulins [88]. There have been several polypeptides that have been fused [89] to increase circulation half-life by exploiting recycling and increased solution size mechanisms. Although half-life extension is possible, there have been several clinical development failures or withdrawals [90].

One of the most successful strategies for developing long-lasting biotherapeutics is the use of non-covalent HSA-binding ligands that are conjugated to the therapeutic polypeptide of interest [91,92]. Since albumin is the predominant protein in the body, weakly associating a modified polypeptide therapeutic results in extending the half-life compared to the unmodified polypeptide. Modification of polypeptides with a fatty acid chain to bind to HSA has been most successful clinically for insulin and glucagon-like peptide (GLP)-1 agonists to treat diabetes, e.g., Tresiba^®^ (insulin degludec), Levemir^®^ (insulin determir), Ozempic^®^ (semaglutide) and Victoza^®^/Saxenda^®^ (liraglutide) [67,93].

An emerging technology is coupling drugs to carriers, such as erythrocytes/red blood cells (RBCs), which has been reported as a unique carrier for drug delivery [94,95] of peptides, receptors and antibodies [96,97]. RBC conjugation can avoid complications associated with transfusion of RBCs and ex vivo manipulations [98]; and can help alter pharmacokinetic profiles [99]. RBC-encapsulated asparaginase is currently in Phase III clinical trials [99].

### 3.2. Depot Formulation Strategies to Prolong Duration of Action

#### 3.2.1. Particulate Formulations

Particulate-associated formulations, such as nanoparticles, microparticles and polymeric implants, have garnered attention for decades as suitable protein drug delivery vehicles. Particulate-associated formulations have also been investigated in efforts to increase circulation times and intracellular targeting. Nanoparticles have been described for the delivery of small molecules, proteins, peptides, vaccines and vaccine adjuvants [100]. Various efforts have been made to improve polypeptide stability, bioavailability and release profiles from particulate-associated formulations and dosage forms, but only peptide formulations have been successfully translated. Often organic solvents, surfactants or high shear processes are required that disrupt protein tertiary structure. Peptide-loaded microparticles that are clinically registered include exenatide (GLP-1 receptor agonist), Sandostatin^®^ (ocreotide) and Lupron^®^ (leuprolide acetate). Details of protein-loaded nanoparticles, which are in early stages of development, are reported in many comprehensive reviews including many critical reviews and commentaries that have sought to analyse the true clinical potential for particle-associated medicines [100,101].

Many polymers have been used to evaluate particulate associated strategies including poly-amino acids (e.g., polylysine), poly-esters (e.g., poly(lactic-co-glycolic) acid (PLGA) [102,103,104,105], polylactic acid (PLA) [106] and poly(ester amide)) [107], polycaprolactone (PCL) [105,108,109], polyanhydrides (e.g., poly(carboxyphenoxy propane-co-sebacic acid)) [110] and carbohydrates (e.g., cyclodextrin [111]). Most of these polymers are not water-soluble necessitating the use of organic solvents during particle fabrication, which is typically deleterious for proteins, but generally acceptable for peptides. PLGA is one of the most commonly used polymers and is a good exemplar that many polymer properties, such as biodegradability, crystallinity and polymer hydrophilicity dependent monomer ratios and polymer molecular weight characteristic, will influence drug loading and release properties [112,113]. As peptides are water soluble, there is a burst release with PLGA formulations, which must be optimised to avoid toxicity and to increase the duration of drug release [15].

Particulate fabrication processes are critical for ensuring reproducibility, maintaining sterility and for consideration of scalability. Many methods have been described to fabricate particulate formulations, including nanoprecipitation [114,115], emulsification-based methods [116], microfluidics, lithography, spray drying, and electrodynamic atomisation (EHDA). EHDA processes are versatile, “top-down” and simple processing techniques that exploit electrical energy to generate solid structures from solutions [117,118]. The processes of producing fibres and particles are termed electrospinning and electrospraying, respectively.

Nanoprecipitation is the most commonly used method for nanoparticle formulation, being simple and requiring a low-energy input. It is based on a reduction in the quality of parent solvent in which the solutes are dissolved to induce precipitation. Precipitation can be achieved by varying the pH, salt concentration or solubility, or after the addition of a non-solvent [114]. However, with the nanoprecipitation method, it can sometimes be challenging to eliminate the solvent, and highly water-soluble drugs are hard to incorporate into the polymer matrix [119].

Emulsions can be oil-in-water (o/w), water-in-oil (w/o) or can even be doubled, e.g., water-in-oil-in-water (w/o/w). Emulsion require kinetic input to disperse one immiscible phase in another [6]. This biphasic system is thermodynamically unstable and requires an emulsifying agent to stabilise the formulation and prevent phase separation [120]. Reversible (such as flocculation, creaming and sedimentation) or irreversible processes (such as Ostwald ripening) can occur over time if there is droplet migration [121]. Emulsions have been instrumental in vaccine formulation. For instance, MF59™, a squalene-based licenced adjuvant, has been used widely in influenza vaccines and other products [100,122].

The use of microfluidic based systems shows promise for designing complex formulation. Microfluidics involves the study, manipulation and utilisation of fluid behaviour—the confined liquid travels through micro-channels and chambers to help control the shape and size of particles, even in sub-picolitre volumes [123,124]. Formation of particles is chiefly dependent on factors such as surface tension, energy dissipation, fluid resistance, laminar flow and viscosity. Scalability and cost of production remain a challenge, and damage to the microfluidic channels by solvents or occlusion can halt the entire production process [125,126]. Another major challenge in making microfluidic systems is the development of devices that can operate over long time periods in a reliable and controllable manner. These systems are, however, also expected to provide benefits in future patient care and possibly fill the gap between animal studies and clinical trials [127]. Micro-channel and extrusion systems can be used to encapsulate polypeptide in core/shell structures [126].

Lithography enables the precise fabrication of 2D and 3D structures on small-scale moulds similar to planographic printing (printing from a flat surface, either on a plate or stone) [128,129]. There are advanced lithography-based methods that include photolithography, moulding technology and particle replication in non-wetting template (PRINT) with the potential to develop biotherapeutic formulations. Photolithography involves the use of light to transfer patterns onto a substrate. PRINT is a highly versatile method with an elastomeric mould containing wells and cavities that can be used to encapsulate oligonucleotides, polypeptides and synthetic viruses to yield an array of sizes, shapes, compositions and surface properties [128,130,131]. Galloway et al. [132] developed a PLGA-based trivalent vaccine against influenza using PRINT technology and compared the generated immune response to soluble antigen. They found that the cylindrical-shaped particles induced significantly higher antibody production in rabbits than the soluble antigen [132]. Soft lithography (use of elastomeric materials) is also used to create channels on chips used in microfluidics and microneedles (micron-sized needles arranged on a small patch) [133] or drug delivery.

Spray drying is a well-established method that utilises a gaseous hot drying medium [134]. Upon nebulisation of a suspension or emulsion, the liquid phase is rapidly evaporated by hot air (or nitrogen) at a high temperature (150–300 °C) to yield particles [114,134]. Parameters, such as the type of atomiser and feed rate can alter the size and morphology of the resulting particles (solid or porous) and also ensure particle components maintain a low enough temperature to avoid active pharmaceutical ingredient (API) degradation by ensuring rapid evaporation and passage through the hot air. Spray drying can be scaled, is cost effective, and can be optimised to be a single step process [135]. However, yield tends to be highly dependent on the scale of the process [135]. In addition, spray drying an aqueous solution of protein can result in loss of conformational stability, leading to potential aggregation and loss of activity which can be mitigated by excipients [136], which sometimes requires the use of mixed aqueous-organic liquids [137].

#### 3.2.2. Gels

Wichterlie and Lim first proposed the use of gelling for the delivery of polypeptides in 1960 [138]. Hydrogels are materials that consist of 3D cross-linked networks with 50–90% by weight of water, and do not flow upon inversion [139]. Gels have been used as wound dressings, vitreous substitutes and regenerative medicine [140]. Polymers commonly used in these systems include PLGA, PEG, poly(vinylpyrrolidone) (PVP), hyaluronic acid or hyaluronan (HA), poly(acrylamide) and collagen [141], as well as natural polymers, such as chitosan, xanthan gum, guar gum and carrageenan [142]. Drug release and degradation mechanisms are dependent on matrix characteristics such as mesh size, mechanical strength and interactions with the protein cargo [143].

Stimuli-responsive hydrogels undergo physicochemical changes in response to different environmental conditions such as pH, temperature and light, although none have been clinically approved yet. Gels can be injected and then undergo a physicochemical change, such as becoming insoluble (i.e., polymer collapse) so as to act as a depot.

Photosensitive hydrogels utilise photosensitive crosslinkers, and the presence of the polymers in the hydrogels allows the use of a laser or alternative external light source to induce gelation in these systems. However, tissue cytotoxicity and limited penetration of light through tissues may limit the use of ultraviolet (UV) light [144]. As a result, near-infrared (IR) light has been investigated as an alternative as it is less absorbed by tissues [144].

Thermosensitive hydrogels are sought to exploit differences between room and body temperature to transition from aqueous solutions to insoluble gels at temperatures above their lower critical solution temperature (LCST) [145]. The temperature-induced change is governed by the balance between hydrophobic and hydrophilic moieties within the polymer in the formulation [146]. The difference in temperature inside and outside the body is typically large enough to induce the crosslinking of the gels as well as trigger drug release [140], but the rate limiting factor to drug release is thermal diffusion [144]. pH-responsive hydrogels are amongst the most studied stimuli-sensitive gels used in pharmaceutical research. Swelling in these hydrogels is controlled by several factors, such as ionic charge, polymer concentration, degree of ionisation and the pH of the swelling medium [16].

The main strategies to prepare drug-loaded hydrogels include drug imbibition, in-situ polymerisation or crosslinking, and two-phase partitioning. Imbibition involves swelling the gel in a solution of free drug. One of the limitations with this approach is that the drug loading levels in the gel are low and are often less than 0.1% weight for proteins. Macromolecules do not readily mix and attempting to mix a protein into a cross-linked macromolecular gel is even more challenging. This limitation can be overcome to some extent if a dehydrated gel network is allowed to swell in a non-aqueous protein solution, dried and then reswollen in water. However, this approach is generally not possible or scalable due to protein denaturation and aggregation.

The in-situ crosslinking and polymerisation approach involves mixing the drug with monomers, crosslinker and initiator, and then allowing the polymerisation reaction to occur. This strategy allows the entrapment of the drug within the hydrogel network; however, orthogonal reaction conditions must be utilised to avoid side reactions between the polymer network and the polypeptide. It is also necessary to (i) remove the leachable initiator, monomer and/or crosslinker reagents and (ii) avoid denaturation and aggregation of the protein during the reaction process [147].

Liquid two-phase systems have been used to partition proteins from cell debris [148]. It is possible to partition a protein from a polymer solution (e.g., poly(ethylene oxide)) where the protein has a lower solubility than the gel (e.g., dextran) [149]. When proteins are absorbed and concentrated in the gel phase, the protein can be released by sequentially contracting the gel with different leaching concentrations. In addition, pH/temperature-sensitive gels can be used to absorb a protein solution. A change in the environment (e.g., pH or temperature) of the gel would cause expulsion of the absorbed solution after separation of the gel from the solution. As a result, the gel and protein solution can be concentrated in a single step avoiding the use of a leaching solution and a drying step [148,149].

### 3.3. Targeting Tissue Components

#### Affinity-Based Drug Delivery

Affinity-based drug delivery systems (DDSs) may avoid common challenges faced during the preparation of controlled release systems [150]. Affinity-based DDS is often based on interactions between an API and its dosage form but can also be related to the interaction of an API with a tissue. Affinity-based DDS is not solely dependent on the properties of the polymer matrix (e.g., pore size and degradation rate) [151], complicated manufacturing steps and surgical administration [152]. A polymer formulation can be modified to give a higher affinity towards drug molecules to increase the drug loading and hence, prolong the drug duration of action [153]. An example of an affinity-based delivery system is the heparin-based delivery system, which releases heparin-binding growth factor, e.g., basic fibroblast growth factor (bFGF) [154].

Much preclinical research has been described for molecularly imprinted polymers (MIPs) [151,155,156]. Typically in MIP systems, polymeric constructs are designed with the target protein acting as a template; removal of the template then creates a free site for further association of the protein [157]. MIPs have been shown to significantly increase drug loading in comparison to non-imprinted systems, as well as slowing release [158]; they have also been described as having stability and durability against harsh conditions [159]. However, some challenges do exist for MIPs, such as an initial burst release with hydrogel MIP systems (due to swelling behaviour) and a lack of sufficient in vivo studies [159].

An alternative approach to achieving affinity DDS is to exploit the specificity of aptamers. Aptamers are short strands of deoxyribonucleic acid (DNA) or ribonucleic acid (RNA) molecules which are capable of exhibiting high affinity towards a target protein [160]. Soontornworajit et al. [161] demonstrated the ability of aptamer-containing hydrogels to markedly slow down the release of platelet-derived growth factor (PDGF), owing to aptamer–protein interactions [161]. Such a system provides a remarkable means of modulating drug release by altering the aptamer sequence to alter affinity. However, while the aptamers for PDGF have been reported, deriving the appropriate sequence for a given protein can be challenging. Systematic evolution of ligands by exponential enrichment (SELEX), the most widely used method for developing such aptamers, can be a tedious and challenging process [160]. Nonetheless, once the aptamer for a given protein is known, this can lead to a powerful strategy to achieve controlled release.

## 4. Polypeptide Delivery

### 4.1. Routes of Administration

Therapeutic proteins and peptides are typically administered parenterally, most commonly via intravenous (IV), SC or intramuscular (IM) routes [162,163]. However, significant research efforts have been directed towards the exploration of other routes of administration, due to the relative invasiveness of parenteral routes, such as the mucosal and oral routes [164,165,166,167,168]. In addition, where local activity or targeted delivery is required (e.g., intravitreal therapy) [162] or blood–tissue barriers exist (e.g., blood–ocular and blood–brain barriers), other routes and modes of delivery beyond the parenteral may be required, especially for sustained drug release and convenience purposes. This section will describe the challenges and formulation strategies for IV, SC, oral/mucosal, ocular delivery, and also highlight the brain as a target for polypeptide delivery (Table 1).

#### 4.1.1. IV Drug Delivery

##### General Concepts and Challenges

IV delivered medicines may be given as a bolus, but proteins are often administered as an infusion [169]. Mini-pumps can allow for the bolus administration of protein therapeutics, and can achieve a precise control of the administration process [170]. The IV route confers the advantage of 100% systemic bioavailability due to the abrogation of first-pass metabolism. Such high availability in comparison to other routes has particularly led to this administration route being used with many antibody-based therapies, which are typically required at high doses (100s of milligrams) [169]. With antibodies, the presence of FcRn means a long half-life is attained upon IV administration [171], whereas peptides (which are susceptible to enzymatic degradation and rapid renal clearance) have a very short half-life upon IV administration [172].

Despite the high bioavailability offered, there are a number of problems associated with the IV route. IV administration is invasive, which can be painful and also presents venous access challenges [173]. Due to the high doses required, IV administration of antibodies is usually by infusion, which necessitates hospital visits and treatment being carried out by trained healthcare professionals. As a result, the overall cost of IV administered medicines can be high [174]. Sterility is another crucial parameter to consider prior to administration, which in turn can increase the cost of manufacture, and steps, such as sterile filtration of a polypeptide can alter its stability [175].

##### IV Formulation Strategies

Although PEGylation remains the most widely applied strategy for formulation biologics for IV administration, alternative conjugation techniques to PEGylation have been explored to prolong the delivery of biotherapeutics given by the IV route. Villalonga and co-workers have reported the preparation of dextran-catalase conjugates by using a 5 kDa dextran coupled to glutamic and aspartic acid residues on the enzyme, and with dextran activated with an ε-aminocaproic acid spacer [176]. A 20-fold (from 0.8 to 16 h) and 7-fold prolongation of half-life (from 0.7 to 5.1 h) were observed, respectively, with corresponding decrease in clearance and increase in total drug exposure [176,177]. A HESylated analogue of anakinra, a recombinant human IL-1 receptor antagonist approved for the treatment of rheumatoid arthritis, has been reported [178]. IV injections to rats showed an enhanced pharmacokinetic profile of the drug with an increased half-life (6.5 times) and a marked decrease in clearance and volume of distribution, indicating reduced diffusion out of the blood compartment [178].

Lindhout et al. [179] described PSA conjugation to α1-antitrypsin for the treatment of antitrypsin deficiency. The conjugate exhibited a terminal half-life of 27 h compared to a 5 h half-life for unmodified α1-antitrypsin and an 18-fold greater bioavailability [179]. Linkage to XTEN led to an increase in the plasma half-life of clotting factor VIIa (FVIIa) in haemophilia A mice models (8.9 h versus 1.2 h) [86], while a terminal half-life prolongation for FIX was noted (from 15 to 40 h). Aghaabdollahian et al. [180] showed a 5-fold increased terminal half-life of a single IV dose of Adnectin-C-PAS compared to the unmodified protein in female BALB/c mice [180]. Schlapschy and co-workers [181] designed polypeptides with repeating glycine-serine units (HAPylation) to increase the hydrodynamic radius and half-life of an anti-human epidermal growth factor 2 (HER2) antibody fragment (Fab), noting a moderate extension in half-life and doubling in hydrodynamic radius [181].

Elastin is an abundant extracellular fibrous protein, which has inspired the use of elastin-like polypeptides (ELPs) for drug delivery. The most commonly used repetitive sequence is the pentapeptide VPGXG, where V (valine), P (proline) and G (glycine) are the short amino acids, and X represents any amino acid excluding proline [182,183]. The large hydrodynamic radius and reduced kidney elimination offered by ELPs [184] help to enhance the half-life of conjugated proteins [183]. In humanised tumour necrosis factor (TNF) rodent models, an anti-TNF-α V_H_H antibody fused with ELP showed a markedly increased half-life of the antibody (24-fold longer than the non-ELPylated form) [184].

Zorzi and co-workers [185] developed an acylated heptapeptide that binds to HSA with high affinity. Conjugation of the ligand to a peptide FXII inhibitor extended the half-life of the peptide from 13 min to over 5 h following IV administration, with coagulation inhibition maintained in rabbits for 8 h [185]. Koehler et al. [186] coupled a naphthalene acyl sulfonamide tag to a peptide targeting FVIIa, which exhibited a 4-fold increased half-life with similar affinity for FVIIa to the peptide alone [186]. Another fusion strategy involves the fusion of insulin to a fluorenylmethyloxycarbonyl derivative of 16-sulfanyl hexadecenoic acid demonstrated by Sasson et al. [187]. They reported a 5-fold increase in plasma half-life of insulin after IV administration in rat models [187].

Stable variable heavy-chain domains (nanobodies) have also been developed as albumin binders [188], and have been employed for increasing the half-life of proteins or peptides in blood [189]. In one study by Tijink et al. [190], a bivalent α-EGFR nanobody was fused to an albumin-nanobody, resulting in an increased serum half-life [190]. Another strategy employing non-covalent albumin interactions is the use of DARPins, which are protein scaffolds derived from naturally occurring ankyrin proteins [191], one of the most abundant binding proteins of the human genome [192]. DARPins exhibit high thermal stability [191] and solubility, and can be expressed in milligrams or gram quantities with a high degree of purity [192]. A study by Steiner et al. [193] reported half-life prolongation of more than 300-fold in cynomolgus monkeys for a model protein fused to a DARPin compared to drug alone (from 2.6 to 20 days), indicating the value of DARPins in generating multifunctional drugs with extended half-lives [193].

Holt et al. [194] isolated stable human anti-albumin single domain antibody fragments (AlbudAbs) with different degrees of affinity and cross-reactivity. This study revealed that these antibodies have similar half-lives to albumin itself [194]. An IL-1 receptor antagonist fused to AlbudAb showed a 130-fold longer half-life in vivo compared to the unmodified antagonist when tested in a mouse model of arthritis [194]. Other applications of the AlbudAb technology have also been reported for IFN-α2 [195] and a mouse anti-TNF receptor 1 antagonist [196]. An AlbudAb fusion to exendin-4 has been tested in phase I human trials by GlaxoSmithKline (GSK), who now own the AlbudAb technology [197,198]. In addition to a prolonged pharmacokinetic profile, the AlbudAb-exendin-4 fusion molecule showed a reduction in postprandial insulin and glucose, and the in vivo half-life was increased 58- to 96-fold in healthy patients [197].

#### 4.1.2. SC Drug Delivery

##### General Concepts and Challenges

The susceptibility of biotherapeutics to degradation and poor oral bioavailability has led to the systemic approach being the principal administration route employed. The SC route provides an alternative to IV administration for many polypeptides and proteins while also avoiding first pass metabolism. In addition, the SC approach could permit the self-administration of biotherapeutics, freeing up healthcare practitioners’ time to focus on issues beyond administration. Consequently, improved patient preference and adherence is reported for SC-administered therapeutics, leading to an overall reduced cost. There are over 60 peptides [11] and 70 antibodies [199] that are clinically approved. For antibodies, reducing injection frequency remains a challenge, with yet any clinically approved antibody tackling this specific challenge. Peptides, as already mentioned, tend to have a short half-life, for instance, the native GLP-1 agonist has a half-life of 2 min, while the first GLP-1 drug has a half-life of 2 h [200]. Such short half-lives mean there is ample scope for the development of formulations, which are capable of extending the duration of action.

The SC tissue is composed of the areolar tissue, playing as a primarily connective role, as well the adipose tissue (Figure 2). The latter, rich in fatty cells, acts as an energy store while also containing fibroblasts, which secrete constituents of the ECM [201,202]. The ECM is composed of glycosaminoglycans (GAGs), such as HA and chondroitin sulphate (CS), which are intricately linked electrostatically with collagen, playing a structural function in the SC tissue. The propensity of HA to swell is well documented and findings indicate that HA possesses 10 times the volume potential of collagen [203,204]. This potential for swelling and thus volume exclusion limits the injectable volume within the SC tissue. The suggested maximum SC injection volume is 2.0 mL [204,205]; higher volumes result in injection site pain linked to rapid hydrostatic pressure changes [206]. Therefore, the SC space is ideally suited for biotherapeutics with high concentration, low volume, or both.

Despite the advantages of SC-administered proteins, there exist a number of challenges with this route. The first is that it still represents an invasive approach to drug delivery. As such, a certain level of “know-how” is required by the patient for them to take their medicine safely, much more than with orally administered formulations. The presence of needles within the delivery device can result in pain during administration, although there has been an improvement in delivery devices with much finer needles (30 G) being routinely used to reduce pain. Modern technologies could completely obviate needles, using instead a needle-free system to deliver a payload of the drug with less associated pain [207]. The SC tissue limits injection volume to approximately 2.0 mL. While such a volume is typically sufficient for the administration of peptides (owing to their potency), it presents a challenge for proteins, which are often required at higher concentrations. At such high concentrations required for efficacy, certain antibodies demonstrate an exponential increase in viscosity [208]. Recent attempts to circumvent this challenge have exploited the unique rapid turnover of HA (ca. 15 h) with the injection of human hyaluronidase (rHuPH20) into the SC space allowing volumes of 5.0–10.0 mL to be feasibly injected [205].

The bioavailability of SC-administered medicines typically trails behind the IV route. Reports have shown wide variations in bioavailability in humans (30–100%) [202]. This reduction in bioavailability is still poorly understood; however, it has been attributed to catabolism at the site of injection [209]. In comparison to the IV route, SC-administered drugs often show higher immunogenicity [210]. This may be particularly the case for larger sized molecules (>16 kDa) which preferentially traverse the lymphatic route [211], rich with immune cells, before entering the systemic circulation [210]. Reports have, however, shown instances of there being no difference in immunogenicity between the SC and IV routes [205].

##### SC Formulation Strategies

In clinically approved peptide products, there are two major strategies to extending the duration of action; the first involves chemical conjugation or biosynthetic fusion of fatty acid, PEG, albumin [212], XTEN [213], elastin-like-polypeptide [214] or Fc region [215] to the peptide. While this chemical conjugation has shown promising clinical results via the SC route, it requires drug re-synthesis, which can lead to loss in higher order structure and unpredictable effects on potency [216]. A second approach relies on carrier mechanisms to deliver a fixed dosage of the drug in its original form over an extended period of time, without influencing the half-life of the therapeutic. Here, the need for re-synthesis is precluded. The hypodermis provides a suitable lodging site for the carrier system, which slowly releases the drug and is then ideally biodegraded.

The overwhelming majority of non-conjugation approaches to depot formulations are based on PLGA microsphere technology. This is owing to the extensive characterisation of the polymer and multiple clinical approvals. The technology works by encapsulating a payload of the drug in the PLGA carrier; the former is then slowly released when surface and bulk erosion of the polymer occurs. As a result, it is possible to achieve a weekly to monthly injection frequency (Table 2). By formulating exenatide in PLGA microparticles (Bydureon^®^), it has been possible to reduce the injection frequency from twice daily to once weekly [217]. Eligard^®^ presents a slightly different approach to the use of PLGA, where the polymer is dissolved in a biocompatible solvent N-methyl-2-pyrrolidone (NMP) and is mixed with the drug compartment just before administration [218]. Once an in-situ implant is administered, a subsequent dosing frequency of up to 6 months is recommended. Zoladex^®^ is a formulation of goserelin dispersed in a PLGA matrix, which is shaped into injectable cylinders, providing up to 3 months of release when administered [219,220].

Insulin represents a major class of drug which has been heavily investigated for modulated release. Glargine, which is formulated at pH 4 to be soluble, functions by precipitating into a depot upon SC injection due to the resultant pH change (to pH 7.2) in the SC tissue. It then slowly releases the drug over 24 h [221]. Another insulin NPH is formulated with zinc (Zn[2]^+^) and protamine as a suspension. The drug is slowly released from the crystalline suspension over a period of 18 to 24 h after SC injection [222]. These formulations, however, are still relatively short-acting, requiring daily injection. A similar Zn[2]^+^ approach was utilised for Taspoglutide, which precipitates upon injection and was projected to allow monthly administration. However, the manufacturers have withdrawn the drug from clinical development, due to unacceptable levels of injection-site and systemic allergic reactions [223].

One noteworthy technology that involves conjugation is ELP [214]. Interestingly, this peptide combination demonstrates LCST behaviour, resulting in the formation of a depot due to the temperature change from room to SC (body) temperature upon injection [224]. The drug is then slowly released from the insoluble coacervate generated, controlling blood glucose for 5 days [214]. Owing to their biocompatibility, multiple studies have examined hydrogel-based formulations, which offer tuneable characteristics including LCST [143,225] for extending the duration of action of biopharmaceuticals. However, there is yet be a clinically approved product for SC administration.

As depicted in Table 2, peptides remain the primary focus for implants to extend the release of biotherapeutics. The application of such technology for antibodies remains absent clinically, but has generated much research interest [225]. Controlled release of immunoglobulin over a period of weeks has been demonstrated with microspheres [226] and thermoresponsive hydrogels [225]. The challenge remains that the generally high concentrations required for SC-administered antibodies [227] are likely to cause aggregation within a delivery system, especially one expected to deliver the antibody over a period of months. Thus, the success of implants as antibody carriers will be dependent on having addressed this issue along with foreign body responses that may arise.

#### 4.1.3. Oral and Mucosal Drug Delivery

##### General Concepts and Challenges

Oral drug administration is a major delivery route for small molecule drugs, and drugs for oral use are currently the most prominent in the market [102]. According to the Food Drug and Administration (FDA), 50% of the 59 new drug approvals in 2018 were for oral use [102]. In addition to its prominence in the therapeutic landscape, the oral route is also the most desirable method of drug administration due to much greater patient acceptance in comparison to the parenteral route [105]. Understanding the mechanisms of oral drug delivery has been a long-standing goal for the evaluation of pharmacokinetics and pharmacodynamics for medicines and potential formulations [228].

The use of the oral route for the delivery of proteins and peptides remains challenging due to the nature of the gastrointestinal tract (GIT) causing generally poor oral bioavailability, as well as leading to proteolytic degradation in the stomach and reduced intestinal absorption [229]. With an oral bioavailability of less than 1% (or less than 0.1% in some cases) [230] and circulation half-lives on the order of minutes to hours [231], a need arises for the development of novel systems to enable delivery of proteins and peptides orally [102]. For example, so far, a formulation of semaglutide (Ozempic^®^), a GLP-1 receptor agonist, is the only oral long-acting medication for the treatment of type II diabetes developed by Novo Nordisk [232]. This is possibly due to its long half-life (160 h) as a result of modified acylation (permitting albumin binding) and the substitution of glycine for aminoisobutyric acid at position 8 of the peptide chain (to avoid enzymatic degradation) compared to liraglutide (Victoza^®^) [233]. Several approaches have been investigated to improve the oral bioavailability of macromolecules.

Structural and biochemical barriers within the GIT prohibit protein and peptide absorption, and limit overall drug bioavailability following oral administration of polypeptides [102,168]. The major structural barrier is the mucus lining of the stomach and intestine (Figure 3), which limits absorption. In addition, pH and enzymes act as biochemical barriers to protein stability due to denaturation [47,102,105,168]. The glycocalyx and mucus of the intestine may also cause protein degradation [234]. In addition to these, the large molecular weights of protein drugs limit drug absorption through the intestinal lining [105]. The oral route is also significantly limited by poor targeting [235]. Further technical challenges exist in the development of oral delivery systems for biologics. Issues regarding scale-up processes, addressing problems associated with traversing biological barriers, and product commercialisation further hinder the development of protein and peptide oral delivery systems [168].

##### Oral and Mucosal-Formulation Strategies

Several formulation strategies have emerged for the delivery of proteins orally (Table 3). The use of protease inhibitors to dampen proteolytic enzyme activity, and permeation enhancers [105,236,237], ultrasound [238,239], microjet systems [238,240] and microneedle capsules [238,241] to improve drug absorption have recently been explored as techniques to maintain the bioactivity of the administered proteins as well as enhance transport through disrupted tight junctions [105]. These formulations are capable of remaining intact throughout transit and allow for drug absorption into the systemic circulation. For prolonged drug delivery purposes, an ideal delivery system should, in addition to the above, have a long residence time in the gut [168] to permit controlled drug release over an appropriate time period. To achieve this, mucoadhesion is one concept of choice for most delivery devices designed for oral drug delivery of biologics [102,168].

Patches are relatively novel systems for oral delivery of small and macromolecules with mucoadhesive properties [242]. Similar to a transdermal patch, intestinal patches are able to encapsulate and protect the desired drug while adhering to the intestinal wall [238,242], thus positioning the drug at the desired absorption site [241]. A patch consists of three layers: a mucoadhesive material, a drug-loaded layer and a relatively impermeable drug-protecting membrane [242]. The patches are then loaded into enteric-coated capsules, for example, which dissolve in the intestine to release the patch [243]. Patches are potentially attractive delivery systems for oral biologics as they are able to prevent proteolysis of encapsulated proteins and peptides, and promote absorption through the intestine by forming a local drug depot with controlled, unidirectional drug release [238]. Patch-based devices have been discussed in previous reviews for the oral delivery of biologics such as bovine serum albumin (BSA) [242], insulin, exenatide [244], human GCSF [245], EPO [246] and calcitonin [247].

Researchers at Intract Pharma have developed the Soteria^®^ technology, a new formulation platform that enables protection of biologics (especially mAbs) against protease enzymes. The technology utilises a synergistic combination of known excipients that, when delivered together with biologics to the colonic lumen, significantly improve enzymatic stability. This allows for high tissue concentrations that previously could not be achieved by simply targeting the biologic to the colon. The technology was developed on the back of findings that immunosuppressive antibodies, such as infliximab and adalimumab, have superior stability in the colonic lumen compared to the stomach and small intestine [248]. Soteria^®^ and Phloral™ [249] platform technologies are currently being investigated for sustained release colon tissue delivery of anti-TNF-α mAbs to provide a novel oral treatment for chronic inflammatory diseases such as inflammatory bowel disease (IBD). The oral gut targeted delivery mechanism can significantly improve therapeutic antibody concentration in the inflamed tissue compared to injection to drive efficacy and potentially reduce the dose.

Polymeric hydrogel systems can be tailored for site-specific and sustained oral delivery of biologics [102], and have become attractive candidates for the controlled release of orally administered protein drugs [105]. For these purposes, hydrogels may either be cationic, anionic [102] or thiol-functionalised [168]. Cationic hydrogels are suitable for drug delivery to the stomach, as they ionise and swell at low pH to release their drug cargo [102]. Anionic hydrogels, on the other hand, exist in a collapsed form at stomach pH (which is lower than the pKa of the hydrogel network). In a low-volume and collapsed state, the hydrogel shields encapsulated biologics from the acidic environment of the stomach and from enzymatic degradation. The hydrogel then swells upon ionisation in the intestine and colon, gradually releasing the drug in a controlled manner [250]. Thiol-functionalised hydrogels are able to attach via disulphide bonds to mucin glycoproteins [168]. A chitosan-based hydrogel for the delivery of insulin has been studied [251], exhibiting increased drug residence in the stomach due to the mucoadhesive properties of the polymer. Anionic hydrogels for the delivery of calcitonin [252], heparin [253], IFN-β [252], and insulin [254] have also been designed with poly(methacrylic acid), alginate, HA and polyacrylic acid [102].

Polymeric particulate carriers are also an attractive delivery system for proteins and peptides via the oral route. The formulations are derived from biodegradable and biocompatible polymers that undergo hydrolysis under certain conditions (pH and other environmental factors) to release the encapsulated drug at the desired location [102]. Systems of this nature are designed either as degradable systems (employing pH-responsive polymers) or mucoadhesive systems (which adhere to the gut wall) [105,168]. Drug release from degradable systems is dependent on the extent of polymer matrix degradation. Ideally the system is expected to remain intact in the stomach, with degradation starting in the small or large intestine [105]. Degradation products from the breakdown of the matrix should also be safely absorbed or excreted. Mucoadhesive particulate systems interact with mucus on the intestinal lining to extend the residence time of a drug and enhance bioavailability [105]. In particular, chitosan-based formulations have been subject to numerous studies owing to their bioadhesive properties [105,263,264]. Mucoadhesive chitosan nanoparticles have successfully been used for the formulation of insulin [265,266] and calcitonin [234] using alginate, dextran sulphate and protease inhibitor liposomes [105]. Although promising, mucoadhesive systems are presented with the unique challenge of mucus turnover (occurring every 50–270 min), which results in a residence time in the intestine of only 4 to 5 h [267].

In addition to mucoadhesive systems, self-emulsifying drug delivery systems (SEDDSs), described as isotropic mixtures of drug, lipids and emulsifiers [268], have been explored by academic and industrial research groups to overcome oral drug delivery barriers and for controlled release of peptides following oral administration [269]. Systems differing from the more traditional SEDDSs such as mucus-penetrating or enzyme-inhibiting SEDDSs have also been generated recently as a means of enhancing permeation through barriers and protecting the drug against degradation by proteases, respectively. SEDDS are, however, susceptible to degradation by pancreatic lipases due to their composition, and thus consideration should be given to the use of excipients, which are less degradable. Solid SEDDSs can also be used for controlled drug release as liquid SEDDSs can be efficiently adsorbed on solid carriers and compressed into tablets using excipients exhibiting prolonged release properties [269]. An oral formulation of cyclosporine developed by Novartis and based on SEDDSs is currently on the market (Sandimmun Neoral^®^), while an insulin SEDDS formulation is in Phase III clinical trials [269,270]. SEDDSs for the delivery of a number of other peptides, including octreotide [258] and leuprolide [259], have also been described by different research groups (Table 3). SEDDSs offer the advantages of being simple to produce with batch uniformity and being cost-effective, making them very promising for oral protein and peptide delivery [269].

#### 4.1.4. Ocular Drug Delivery

##### General Concepts and Challenges

The intraocular delivery route has been exploited to deliver drugs directly to ocular tissue, especially for conditions affecting the back of the eye/posterior segment such as diabetic retinopathy, posterior uveitis and age-related macular degeneration (AMD). Intraocular drug delivery can be achieved via several routes, which may be intracameral (injection directly into the anterior segment of the eye), suprachoroidal (drug delivery to the space between the sclera and the choroid—the suprachoroidal space), intrastromal (drug delivery directly into the corneal stroma, bypassing the corneal epithelium and tear fluid drainage) or intravitreal (injection into the posterior segment of the eye—the vitreous) (Figure 4).

Vascular endothelial growth factor (VEGF) is the key player in the pathogenesis of choroidal neovascularisation (CNV) and tumour angiogenesis, with four different homodimeric proteins or isoforms formed by VEGF messenger RNA (mRNA) splicing, namely, VEGF_121_, VEGF_165_, VEGF_189_ and VEGF_206_ (based on the number of amino acids present). The intravitreal delivery of anti-VEGF medicines has greatly changed the management of AMD and diabetic retinopathy [146]. Most commonly, pegaptanib (Macugen^®^, OSI Pharmaceuticals), aflibercept (Eylea^®^, Regeneron Pharmaceuticals), ranibizumab (Lucentis^®^, Genentech/Novartis) and bevacizumab (Avastin^®^, Genentech-used off-label) have been used in treatment, closely followed by the development of newer molecules, such as brolucizumab (Beovu^®^, Novartis).

Proteins and peptides, despite their benefits, are plagued by short half-lives and thus typically undergo rapid elimination from the ocular space in the order of minutes to several weeks, depending on the route of administration. This results in frequent drug administration being required to maintain drug concentrations at therapeutic levels. For this reason, implants and injectables were initially studied to prolong drug release of proteins (and small molecules) in the eye. However, although clinical success has been achieved with small molecule injectables and implants (a good example of which is the sustained delivery of steroids to the back of the eye), challenges arise in the formulation of longer lasting protein therapeutics due to instability relating to structural and environmental factors such as pH, compatibility with formulation excipients, shear forces and salt concentration [271,272].

Numerous studies have also shown that ADAs are produced in the vitreous in response to the injection of therapeutic proteins into the eye [273,274], which significantly impact the pharmacological properties of injected molecules [275]. ADAs may decrease or, less commonly, increase drug exposure and produce immune response-related toxicologic effects [274]. The VIEW study for bi-monthly intravitreal administration of 2.0 mg/eye of aflibercept was associated with ADAs in a number of patients (1–3%) [276]. Approximately 8–9% of patients treated with Lucentis^®^ developed ADAs after monthly intravitreal dosing; however, there was no reported impact on the pharmacokinetic and pharmacodynamic safety or efficacy of the drug [274]. There have also been a few reports of systemic hypersensitivity to Macugen^®^ upon initial drug exposure [277]. This factor, in addition to ocular tolerability complications associated with intravitreal drug delivery [274,278,279], prohibits the use of frequently administered intravitreal injections of biologics. Complications including sterile endophthalmitis [280,281], intraocular pressure (IOP) elevation [282], ocular inflammation [281], and retinal detachment [283], have been reported following intravitreal anti-VEGF injections of bevacizumab, ranibizumab and pegaptanib [284,285]. There is thus a need for extended or sustained protein and peptide delivery via the intraocular route, leading to the development of several novel formulation approaches capable of slowing drug release to permit less frequent drug dosing and reduce risk to patients. More comprehensive reviews on the challenges on vitreal drug delivery have been reported [286,287].

##### Ocular Formulation Strategies

Drug-loaded implants have been used for several years for the delivery of small molecules to the back of the eye. Many of these implants have been approved by the FDA and have successfully made their way to the clinical setting, including Vitrasert^®^ (ganciclovir), Retisert^®^ (fluocinolone acetonide) and Ozurdex^®^ (dexamethasone). While polymeric implants have thus established their clinical usefulness in the delivery of small molecules, their use in the delivery of polypeptides has been limited so far and they are yet to reach the market.

Genentech has developed a surgically inserted refillable port delivery system (PDS) that is introduced through the sclera for the extended release of ranibizumab in patients with neovascular AMD [288]. Though non-biodegradable, the reservoir does not require removal from the eye. Phase II trial results indicated that in addition to being well tolerated, the PDS has the potential to reduce the treatment burden in AMD by reducing the total number of treatments required by patients [288]. Delivery of the peptide insulin through an acidified absorbable gelatin sponge-based intraocular device (Gelfoam^®^) has been reported [289]. Lee and co-workers achieved great success for systemic insulin delivery via the intraocular route, but it is proposed that this device may potentially find ocular applications via intravitreal delivery to the posterior segment of the eye, with necessary modifications to allow drug release in a sustained manner [290]. Graybug Vision, Inc., has developed injectable PLGA microparticles of sunitinib, a VEGF/PDGF (platelet-derived growth factor) antagonist (GB-102) for wet AMD [291]. NT-501 (Renexus^®^) is an encapsulated cell technology (ECT) developed by the Neurotech Pharmaceuticals and Noah Groups for the treatment of atrophic AMD, glaucoma and retinitis pigmentosa [292]. Ocular implants made from this technology contain a semipermeable hollow fibre matrix of biocompatible polymers (such as collagen and HA) encapsulating genetically modified cells, and are inserted via the pars plana (the part of the ciliary body within the uvea) for attachment to the sclera [292,293]. It is currently in Phase III trials and consists of human retinal pigment epithelium (RPE) cells transfected with plasmids encoding for ciliary neurotrophic factor (CNTF) [292]. The use of ECT for the secretion of soluble VEGFR1 from RPE cells has also been proposed by Kontturi et al. [294] and has been described as a potential formulation technique for the treatment of posterior segment diseases such as AMD, diabetic macular edema (DME) and diabetic retinopathy [295].

Angkawinitwong et al. [296] reported the sustained release of bevacizumab encapsulated in fibres with a PCL sheath via electrospinning, using two different pH values (pH 6.2, F_beva_ and pH 8.3, F_bevaP_) [296]. The in vitro half-life was 11.4 ± 4.4 days and 52.9 ± 14.8 days for F_beva_ and F_bevaP_, respectively [296]. Biodegradable core-shell electrospun nanofibers of bevacizumab were designed by de Souza et al. [297]. The nanofiber shells were synthesised from gelatin and PCL, with a PVA core protecting and storing bevacizumab. No cell toxicity was observed, and the formulation maintained anti-angiogenic properties and displayed a slow release [297].

Micro- and nano-sized delivery systems are extensively studied in the design of formulations for small molecules and proteins. Numerous systems, including microspheres, liposomes, dendrimers, nanowafers and micelles, have been used as carriers for targeted protein and peptide delivery; and are known to release drugs in the vitreous while maintaining concentrations at a therapeutic level for prolonged periods [290]. A number of particulate systems have already been reported to prolong protein drug release following intraocular delivery [1]. Pandit et al. [298] designed chitosan-coated PLGA bevacizumab nanoparticles for delivery into posterior ocular tissue [298]. Extended antibody release was achieved with drug flux reportedly higher than that of the drug alone [298]. The intravitreal delivery of connexin43 mimetic peptide for the reduction in ocular vascular leak and retinal ganglion cell death was also described in a study by Chen et al. [299]. The peptide was encapsulated in PLGA nano- and microparticles for the purpose of promoting ganglion cell survival using a retinal ischaemia-reperfusion model in rats. In vitro release studies showed an initial burst release of drug followed by slow total release and complete particle breakdown after 63 days (nanoparticles) and 112 days (microparticles) [299].

Hydrogels are among the most studied systems for the delivery of both small and large molecular weight drugs to the eye [272]. Hydrogels for intravitreal drug delivery are advantageous because they take up small volumes within the vitreous space and are biodegradable. They are also useful for circumventing the need for frequent repeat injections—which presents several risks and complications to patients—by acting as a drug depot even for unstable protein molecules [140]. There are a number of in-situ gelling systems reported for ocular delivery of proteins [225,300,301,302,303,304,305,306,307,308]. Xue et al. [305] developed thermosensitive gels of anti-VEGF drugs (bevacizumab and aflibercept) using a PEG-polypropylene glycol (PPG)-PCL multiblock copolymer (hydrophilic–hydrophobic biodegradable copolymer) [305]. Zero-order drug release was achieved in vitro over 40 days. Anti-VEGF activity was seen in a neovascularisation rabbit model, human umbilical vein endothelial cell (HUVEC) proliferation in vitro and angiogenesis inhibition of ex vivo choroidal explant [305]. Yu et al. [306] reported the controlled delivery of bevacizumab using an in-situ gel consisting of vinylsulfone-functionalised HA and thiolated dextran [306]. The gel prolonged bevacizumab release with therapeutically relevant concentrations for 6 months in rabbit vitreous [306]. Sustained release of Avastin^®^ (bevacizumab) was achieved with a polysaccharide crosslinked hydrogel of chitosan and oxidised alginate [309], and PEG hydrogels formed via thiol-maleimide reactions [310]. Tyagi and co-workers [307] described the suprachoroidal delivery of bevacizumab from a light activated PCL dimethacrylate (PCM)-hydroxyethyl methacrylate (HEMA) hydrogel [307].

Shear-thinning hydrogels are formed by ex vivo self-assembly through physical interactions such as hydrogen bonding, hydrophobic interactions, and guest–host interactions [311]. These hydrogels become injectable under high mechanical shear and are capable of flowing through a needle, reforming their bonds and forming a cohesive depot at the injection site [312]. Their properties allow injection without needle occlusion, permitting homogenous encapsulation of a drug cargo and recovery of their initial state post-injection, making them suitable for the delivery of proteins [313,314,315,316]. It is essential that these hydrogels self-assemble at physiological conditions, flow freely through a syringe and self-heal after injection [272,312]. The use of shear-thinning gels can overcome issues encountered with the use of in-situ crosslinking systems, such as syringe or needle clogging and reduction in the risk for embolisation upon systemic administration [317]. The use of these hydrogels for small interfering RNA (siRNA) [318] and matrix metalloproteinase (MMP) [313] delivery have been explored. To ensure that shear-thinning hydrogels become clinically relevant for protein delivery to the posterior segment, drug encapsulation and release kinetics must be adequately tuned [272].

Thermoresponsive hydrogels undergo transitions from liquids prior to injection to cohesive gels post injection into the vitreous. They benefit from changes in physical, biological or chemical cues, undergoing physicochemical changes in the presence of even minor changes in their immediate environment [272,319]. They undergo rapid transformation to viscoelastic gels upon injection, prolonging their residence time and sustaining drug release, which will allow for less frequent dosing with enhanced bioavailability and lowered systemic absorption [320]. For intravitreal delivery, they also offer benefits of being transparent and clear (important for unhindered vision), and the ability to protect the incorporated drug from degradation in the vitreous [321]. To design an efficient in-situ gelling system, it is vital that the rate of reaction be rapid enough to encapsulate the protein and prevent the removal of gelling precursors, but sufficiently slow to allow the injection of the pre-gel solution [271]. Awwad and co-workers [225,300,308] have reported the in vitro delivery of antibodies with thermoresponsive hydrogels. These authors [308] reported the sustained release (over 30 days) of bevacizumab and PEG-conjugated ranibizumab (PEG-Fab_rani_) from a crosslinked *N*-isopropylacrylamide (NIPAAM) hydrogel [308]. In another study [300], bevacizumab was released for at least 2 months from a biodegradable NIPAAM-acrylated HA (Ac-HA) hydrogel [300]. Functional bevacizumab was reported throughout the study using enzyme-linked immunosorbent assay (ELISA) [300]. Interpenetrating polymer networks (IPNs), consisting of a blend of two or more polymers, with at least one polymer synthesised in the presence of the other [322], have also been investigated as innovative drug delivery systems for proteins. Egbu et al. [225] reported the delivery of infliximab from an IPN of crosslinked NIPAAM and HA and compared its release with a functionalised HA hydrogel, i.e., HA substituted tyramine (HA-Tyr). The thermoresponsive hydrogel displayed a slower release than HA-Tyr, with an infliximab release of 24.9 ± 0.4% by day 9 [225].

A number of authors [304,323] have studied systems incorporating particulate technology into gels for ocular delivery of proteins. Kang-Mieler and co-workers have developed aflibercept-[304,323] and ranibizumab-loaded microsphere-thermoresponsive hydrogels [301]. They employed PLGA microspheres within an NIPAAM hydrogel for aflibercept and for ranibizumab, a degradable PLGA microsphere system within a PEG-co-(L-lactic acid) diacrylate-NIPAAM (PEG-PLLA-DA/NIPAAm) hydrogel was described, with drug loaded into the microspheres in respective studies. In both studies, protein release was observed for 6 months post injection [304,323]. In vivo studies were also conducted to analyse the safety profile of the aflibercept-loaded system, and showed no abnormalities or significant changes to retinal function [304].

Affinity-based systems have also been explored for prolonged ocular delivery of proteins. Components of the vitreous such as HA [152] and type II collagen [324,325], as well as molecules in the eye such as melanin [326,327] and components of the inner limiting membrane (ILM) [328], have been identified as potential targets for ocular binding affinity. Researchers from Novartis have characterised a hyaluronan-binding peptide (HABP) fused to the heavy chain or Fc region of a number of proteins (Fab, IgG and scFv) for the development of long-acting anti-VEGF drugs (LAVAs). Compared to unmodified protein, these LAVAs showed a 3 to 4-fold increase in half-life in both rabbit and monkey retinal vascular disease models [152]. Natural human proteins, such as RHAMM, LYVE-1 and CD44 have been reported to bind to HA with modest affinity [329]. Ghosh et al. [152] have also described affinity delivery using the link domain of TSG-6 (TNF-stimulated gene 6), which has shown moderate affinity for HA. Studies by the group showed a 3-fold increase in half-life in monkey eyes using a fusion of the HA-binding component of TSG-6 to an anti-VEGF Fab [152,330]. Michael et al. investigated an intravitreally injected VEGF-trap fused to heparan-binding domains (“Sticky-trap”) which interact with heparan sulphate proteoglycans (HSPGs) on the lens, ILM and choroid. Binding to HSPGs promoted protein retention within the eye for 12 days compared to original VEGF-trap [328]. A patent filed by Roche reported the preparation of a multispecific binder, i.e., a recombinant fusion protein (peptidic linker), where the first binding site of the fusion protein specifically binds to a target associated with an eye disease (Fab or scFv), whereas the second binding site specifically binds to type II collagen (scFv). The therapeutics ocular target can target VEGF, PDGF-B, angiopoietin-2 (ANG2) or IL-1β. Fab_coll_ displayed 2.7- and 3.2-times increased diffusion time in phosphate buffered saline (PBS) with collagen and vitreous fluid respectively [325].

### 4.2. Targets

#### 4.2.1. Targeting the Brain

##### General Concepts and Challenges

Drug development to treat brain disorders has greatly increased in the last 2 decades [331]. The continuous advancement in biotechnology and recombinant DNA mechanisms for the synthesis of protein and peptide drugs has led to the development of therapeutics for the treatment of neurological and neurodegenerative disorders such as Parkinson’s disease (PD), amyotrophic lateral sclerosis (ALS), multiple sclerosis (MS), Alzheimer’s disease and Krabbe’s disease [332]. Many neurotrophic factors, such as nerve growth factor (NGF) and brain-derived neurotrophic factor (BDNF), have been investigated for treating these conditions.

Current strategies to target the blood–brain barrier (BBB) involve injection, nasal delivery, and constant infusion with a minipump system or catheter [333,334]. There is, however, a high risk of infection with prolonged use of such invasive systems [335]. Another major challenge in brain drug delivery is associated with the unique and complex environment of the central nervous system (CNS), the BBB (Figure 5), blood–retinal barrier (BRB) and blood–spinal cord barrier (BSCB) [334]. Systemic delivery of proteins to the brain faces two major problems that make diagnosing and treating brain-related diseases very difficult: insufficient drug concentrations passing across the BBB and the rapid serum clearance of the active ingredient [336]. The passage of substances from the blood into the CNS is controlled by the BBB [337]. A drug molecule can normally pass through intercellular junctions (via the paracellular pathway). However, the tight junctions between two opposing cellular membranes limit this passage to small ions and hydrophilic molecules with a hydrodynamic radius <11 A [338] or between 400 and 500 Da [337], making protein transport difficult. Endocytic mechanisms that involve receptor mediated transcytosis (RMT) or adsorptive-mediated transcytosis (AMT) can facilitate the transport of some macromolecules and peptides across the BBB [339]. P-glycoprotein is also responsible for the limited permeation in the CNS [340] especially for large molecules >500 Da [341], and its function can be affected by brain inflammation and oxidative stresses [332].

Glioblastomas are the most common primary brain tumours. The BBB is damaged when the tumour cells disrupt and compromise its integrity. The resultant blood–tumour barrier (BTB) is formed by new blood vessels (brain tumour capillaries) and has a higher permeability than the BBB [341]. The vascular integrity of the BTB is also disrupted during tumour progression [341]. The combination of the BBB and the BTB forms a major barrier for brain tumour drug delivery. Failure of drug therapies arises from poor drug accumulation in the brain tumour site and short half-life of drugs in the tumour due to early dissipation into the cerebrospinal fluid (CSF) and interstitial fluid (ISF) [342]. CNS diseases, such as Alzheimer’s diseases and human immunodeficiency viruses (HIV)-related neuropathologies, can affect functional components of the BBB responsible for the transport of molecules in and out of the brain [332].

##### Brain Formulation Strategies

Any formulation that needs to cross the BBB requires a specific transporter for an active transport mechanism or an appropriate physicochemical property for passive diffusion [338]. Therefore, new technology-based approaches are needed, such as microparticles, functionalised nanocarriers, liposomes and chimeric peptide technology. Currently, the only FDA-approved intracranial drug delivery system is biodegradable discs infused with carmustine (Gliadel), which are placed into the tumour-resection cavity [343,344]. A study by Spencer and Verma [337] targeted the receptors on the BBB by addition of the low-density lipoprotein receptor (LDLR)-binding domain of apolipoprotein B (ApoB) to enable protein delivery to the CNS [337]. A lentivector system was used to deliver the lysosomal enzyme glucocerebrosidase and secreted glucocerebrosidase (sGFP) to the neurons and astrocytes by both IV and intraperitoneal (IP) injections. This system consists of hybrid viral particles composed of RNA from the lentiviral genome, enzymes and core protein from the lentivirus and the envelope protein of another virus [345]. IP delivery was sufficient to transport protein to the CNS across the BBB. The system utilised the liver as a depot organ to reduce the frequency of drug administration [337,346].

Shoichet and coworkers have reported the controlled release of proteins to address brain issues [335,346,347,348], such as stroke [335,346,348] and spinal cord injuries. [347]. Wang et al. [348] reported the preparation and evaluation of recombinant human epidermal growth factor (rhEGF) conjugated to methoxy-PEG-propionaldehyde (mPEG-PPA, 5 kDa) for the development of mono-, di- and tri-PEG conjugates for the treatment of stroke-injured brain via an invasive minipump/catheter system [348]. The hydrodynamic radius was increased by a factor of 1.5 and 1.2 after the addition of the first and second 5 kDa PEG, respectively. PEG modification was seen to reduce the rate of elimination of growth factor by an order of magnitude [346]. Tuladhar et al. [346] reported the delivery of EPO (and cyclosporine) from a composite hydrogel comprising drug-loaded PLGA particles dispersed in HA and methylcellulose (MC) via surgery in a rat model of stroke [346]. A 3-fold drug increase. as compared to systemic delivery, was reported with their formulation, since it was able to pass the BBB [346]. Cooke et al. [335] designed a shear-thinning controlled release vehicle consisting of MC and HA to deliver epidermal growth factor (EGF) and PEG-EGF for the treatment of stroke (via surgery) [335]. PEG-EGF was able to penetrate deeper into the brain than the unmodified EGF, showing at least 2- and 7-fold greater protein accumulation in uninjured and stroke-injured brains, respectively. In addition, PEG-EGF displayed higher stability and greater effectiveness in stimulating neural stem/progenitor cell (NSPC) proliferation. A longer half-life (3.85 h) than EGF alone (1.69 h) was reported in stroke-injured brains [335]. Kang et al. [347] reported the intrathecal delivery of PEG conjugated fibroblast growth factor 2 (PEG-FGF2) from a similar formulation of MC and HA for the treatment of spinal cord injury [347]. The rate of elimination was reduced with PEG-FGF2 (5 kDa PEG maleimide) and a higher concentration (than FGF2) was detected in the spinal cord, due to the ability of the PEGylated system to evade phagocytosis [347].

Several strategies have been introduced to overcome the BTB and improve brain drug delivery by targeting the glioma cells to improve survival time (usually 15 months) [349]. Temozolomide (small molecule) and bevacizumab are FDA-approved drugs for the treatment of glioblastoma [350]. Sousa et al. [349] used the intranasal delivery of bevacizumab-loaded PLGA nanoparticles to cross the BBB for the treatment of glioblastoma multiforme [349]. The nanoparticles displayed higher protein concentrations in the brain and higher exposure over 7 days compared to intranasally administered free drug [349], with a delayed release of bevacizumab [351].

Lampe et al. [352] reported the delivery of neurotrophic factor (NF)-loaded PLGA microparticles entrapped within a PEG-based hydrogel to treat Parkinson’s disease [352]. BDNF and glial-derived neurotrophic factor (GDNF) showed 56 and 28 day release periods respectively, after surgical implantation into rodent brain tissue with a significant decrease in microglia [352]. Huang and co-workers [353] evaluated the delivery of human GDNF (hGDNF)-lactoferrin modified nanoparticles in a rotenone-induced chronic Parkinson’s model, concluding that the formulation can be used as a long-term therapy for the treatment of neurogenerative diseases [353]. A 4-fold increase in expression of reporter genes in the brain was reported after the delivery of these nanoparticles compared to unmodified nanoparticles [353]. Other long-lasting formulations for the treatment of Parkinson’s disease include PLGA microparticles (delivery of VEGF [354] and GDNF) [354,355,356,357,358], PLGA-collagen microparticles (delivery of GDNF fused with collagen binding peptide) [359] and poly(butyl cyanoacrylate) nanoparticles (delivery of nerve growth factor, NGF) [360]. Zhang et al. [361] evaluated lectin-modified PEG-PLGA nanoparticles of bFGF to increase CNS permeability in Alzheimer’s disease rat models via intranasal delivery [361]. The areas under the concentration–time curve of the drug in the nanoparticles following intranasal administration were reported to be from 1.79- to 5.17-fold higher compared to after IV administration, 0.61- to 2.21-fold higher compared to intranasal solution and 0.19- to 1.07-fold higher compared to unmodified bFGF [361].

## 5. Conclusions

Considerable progress has been made in the development of long-acting biologics for local or targeted therapy, despite significant barriers to drug delivery. Novel formulations and processing techniques have contributed to this progress and facilitated clinical translation to the market for a number of proteins and peptides. However, there is still a major unmet need for implantable devices or injectable formulations that can successfully and efficiently reach their targets with their payload intact and provide the required pharmacokinetic profile upon administration. Clinical translatability is another crucial area where advancements are desired. Although small molecule implants have been developed and are commercially available, there are to date no such formulations for proteins especially, despite numerous strategies already described, which have not proceeded to clinical development. Large-scale production and manufacturing of biologics are still challenging for biopharmaceutical companies despite the depth of knowledge in the field. This, and other factors, including designing appropriate systems and developing excipients and materials for prolonged release, are critical steps to be considered. Future efforts should be focused on developing stable and compatible long-lasting formulations of biotherapeutics.

## Figures and Tables

**Figure 1 pharmaceutics-12-00999-f001:**
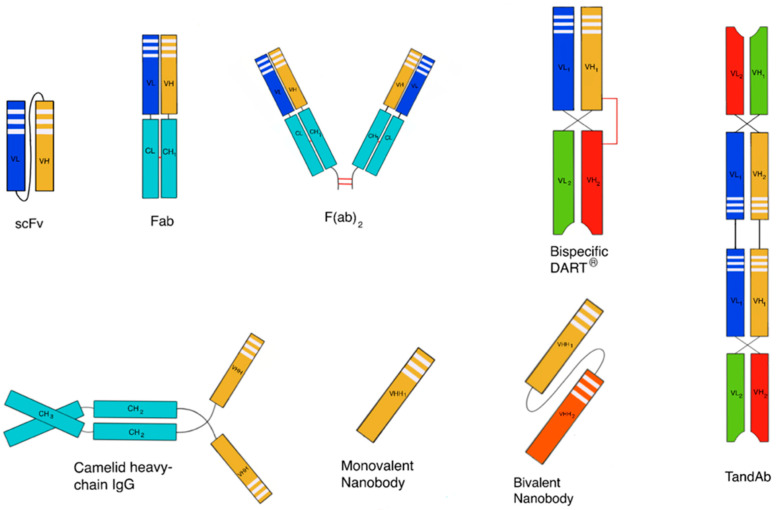
Schematic representation of a few examples of derived fragment technologies from antibody formats. Some of them are used for the development of stable protein scaffolds.

**Figure 2 pharmaceutics-12-00999-f002:**
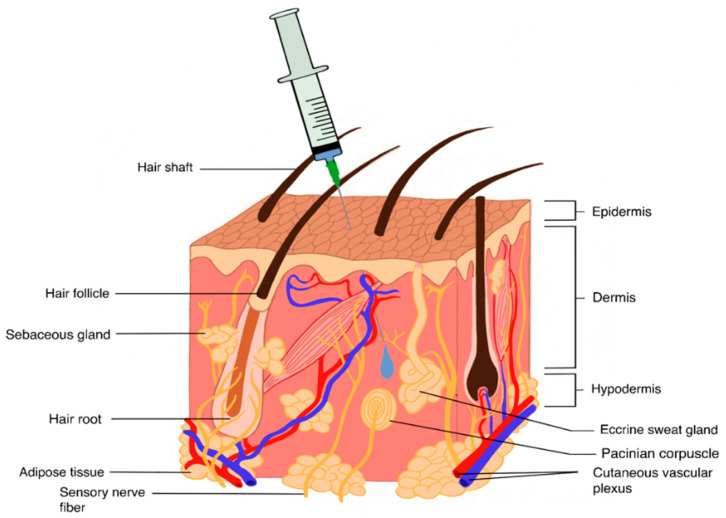
General representation of the SC tissue, which contains a complex structure of proteoglycans and collagen. The injection volume is limited to 2.0 mL; higher volumes can lead to injection site pain due to a rapid build-up of hydrostatic pressure.

**Figure 3 pharmaceutics-12-00999-f003:**
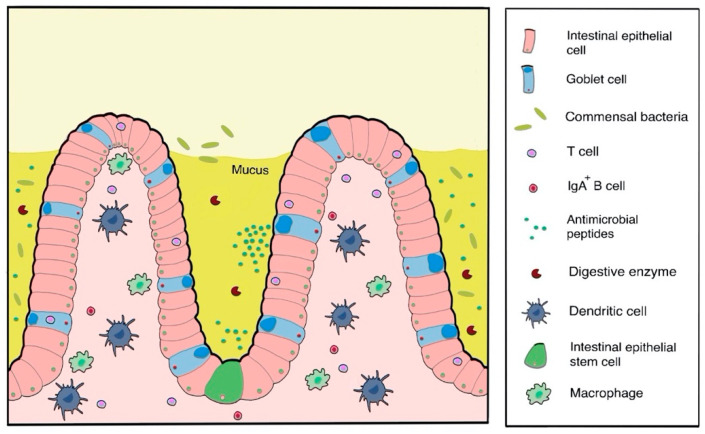
One of the major structural barriers for oral drug delivery is the intestinal lining that comprises of a number of factors, such as digestive enzymes, that can affect the bioavailability of administered polypeptides.

**Figure 4 pharmaceutics-12-00999-f004:**
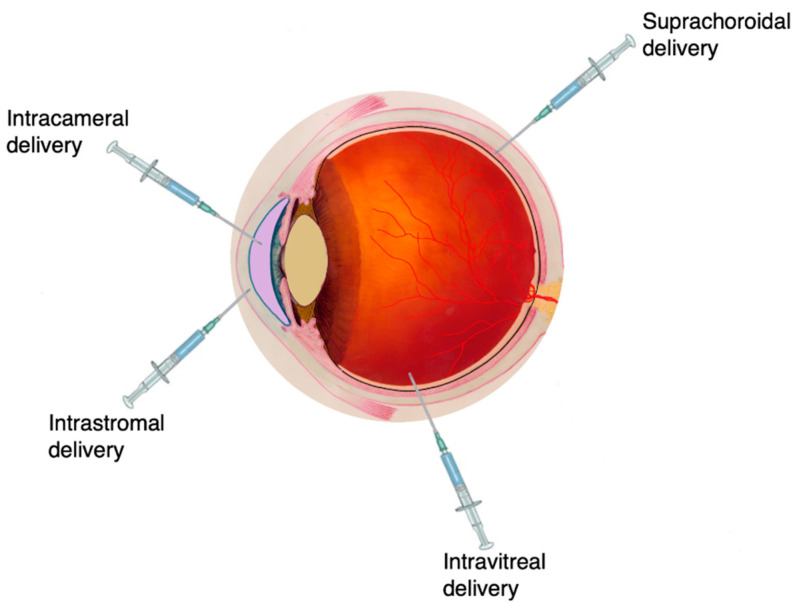
Common techniques to inject medicines to the eye. Administration via intravitreal injection is the most frequently used strategy, especially for targeting the back of the eye, such as the vitreous. Image adapted from National Eye Institute, National Institutes of Health. CC BY 4.0 license.

**Figure 5 pharmaceutics-12-00999-f005:**
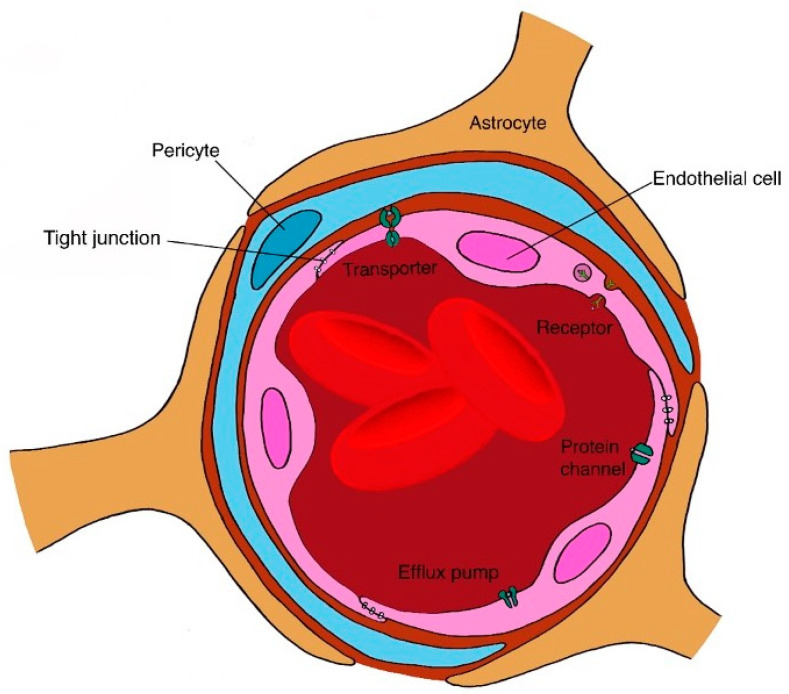
Structure of the BBB, a highly specialised structure formed by a tight monolayer of endothelial cells. The BBB is a common barrier for many formulations especially for large molecules, such as proteins.

**Table 1 pharmaceutics-12-00999-t001:** Common challenges and barriers with selected routes of administration for the delivery of polypeptides including a few examples of formulation strategies for each route.

Route/Target	Examples of Targeted Biomolecules	Barriers to Delivery	Challenges	Examples of Formulation Strategies
**Routes of delivery**				
SC	Insulin, exenatide, insulin growth factor-1, canakinumab somatotropin, mecasermin, omalizumab, etanercept, and IFN-α2b	Injection site catabolism and limited volume in SC space	Reduced bioavailability, drug degradation	Microspheres, thermoresponsive hydrogels, microparticles, conjugation (e.g., polymers) and affinity DDS
Oral	Insulin, octreotide, exenatide, salmon calcitonin, parathyroid hormone, desmopressin, GCSF, erythropoietin, leuprolide and semaglutide	Intestinal mucosa, cellular tight junctions, efflux pumps, pH and enzymes	Drug degradation, decreased absorption and loss of biological activity	Protease inhibitors, absorption enhancers, mucoadhesive systems and polymeric carriers
Ocular	Bevacizumab, aflibercept, brolucizumab, pegaptanib, insulin and ranibizumab	Corneal tight junctions, the sclera, efflux pumps, BRB and ILM	Rapid elimination, short half-life and degradation	Nanoparticles, microparticles, liposomes, microneedles, hydrogels, dendrimers, conjugation (e.g., PEG/other polymers), affinity DDS and micelles
**Target of delivery**				
Brain	Human EGF, FGF-2 and bevacizumab	BBB, CNS, vascular barriers, cellular tight junctions	Short half-life, decreased drug concentration and poor drug distribution	Microparticles, 11 functionalised nanocarriers, conjugation (e.g., PEG) and liposomes

BBB: blood–brain barrier; BRB: blood–retinal barrier; CNS: central nervous system; DDS: drug delivery system; EGF: epidermal growth factor; EPO: erythropoietin; FGF: fibroblast growth factor; FSH: follicle stimulating hormone; GCSF: granulocyte colony-stimulating factor; IFN: interferon; ILM: inner limiting membrane; LHRH: luteinising hormone releasing hormone; PEG: poly(ethylene glycol) and SC: subcutaneous.

**Table 2 pharmaceutics-12-00999-t002:** Clinically approved long-acting biotherapeutics administered via the SC route.

Type	Name	Indication	Strategy	Release Mechanism	Dosing
Leuprolide	Lupron Depot	Prostate cancer	PLGA microparticles	Gradual release from the particulates	Monthly or every 3, 4 or 6 months
	Eligard		In-situ forming PLGA depot	Gradual release from a depot	Monthly
Octreotide	Sandostatin LAR	Acromegaly cancer	PLGA microparticles	Gradual release from the particulates	Monthly
Lanreotide	Somatuline Autogel	Acromegaly cancer	Self-assembly into nanotubes	Release over a long period	Monthly
Goserelin	Zoladex	Prostate cancer	PLGA implant	Gradual release from the implant	Up to 12 weeks administration
Insulin	NPH	T1 diabetes	Formulated with Zn[2]^+^ and protamine	Slowly released from crystalline suspension	18–24 h
	Glargine	T1 and T2 diabetes	Formulation in acidic pH (initially, substitution of α Asn21 with Gly addition of 2 Arg groups to β chain)	Initially soluble formulation precipitates upon injection	20–24 h
GLP-1 agonist	Exenatide (Bydureon^®^)	T2 diabetes	Encapsulation in PLGA microspheres	Release follows polymer degradation	Weekly

Arg: arginine; Asn: asparagine; Gly: glycine, GLP: glucagon-like peptide; NPH: neutral protamine hagedorn; PLGA: poly-lactic-co-glycolic acid and Zn[2]^+^: zinc ion.

**Table 3 pharmaceutics-12-00999-t003:** Formulation strategies for the delivery of polypeptides via the oral route.

Depot System	Therapeutic Biomolecule	Delivery System Design	Example Studies
Patches	Salmon calcitonin, insulin, exenatide	Polymeric matrix system of sodium CMC, carbopol and pectin coated with ethyl cellulose	3-, 13- and 80-fold increase in half-life of calcitonin, insulin and exenatide, respectively [244]
	Insulin	Surfactant-coated chitosan-ethyl cellulose patch system	Increase in local insulin concentration and rapid insulin permeation across intestinal epithelium [255]
	Insulin	Patch-permeation enhancer system in pH-responsive enteric coated capsule	Unidirectional insulin over 3–4 h demonstrated in release studies in rats [243]
Hydrogels	Calcitonin, human growth hormone	Poly(methacrylic acid-grafted-PEG) and poly(methacrylic acid-co-N-vinyl pyrrolidone) hydrogels	Slow drug release at low pH [256]
	Anti-TNF antibody	Poly(methacrylic acid-grafted-PEG) and poly(methacrylic acid-co-N-vinyl pyrrolidone) hydrogel microspheres	Serum antibody levels detectable after 4 h [257]
SEDDs	Leuprolide	PEG-35 castor oil, glycerol monocaprylocaprate, propylene glycol and triglyceride-containing SMEDDs	Sustained drug release (50%) within 30 h [258]
	Octreotide	Oily phase ion-paired complex of octreotide and deoxycholate in SNEDDs	Sustained drug release for at least 24 h [259]
Polymeric and mucoadhesive particulate carriers	Protamine insulin	Mucoadhesive nanoparticles	Increased gut retention time and sustained hypoglycaemic effect in rat models [260]
	Insulin	Chitosan/dextran sulphate nanoparticles	Sustained release over 10 h [257]
	Insulin	EDTA-poly(glutamic acid)-chitosan degradable nanoparticles	Prolonged hypoglycaemic effect by tight junction permeation [261]
	Insulin	HA pH-responsive nanoparticles	Hypoglycaemic effect lasting up to 8 h and drug protection against enzymatic degradation [262]

CMC: carboxymethylcellulose; EDTA: ethylenediaminetetraacetic acid; HA: hyaluronic acid; PEG: poly(ethylene glycol); SEDDS: self-emulsifying drug delivery systems; SMEDDs: self-microemulsifying drug delivery system, SNEDDs: self-nanoemulsifying drug delivery system and TNF: tumour necrosis factor.

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
