# Peer review of "Injectables and Depots to Prolong Drug Action of Proteins and Peptides"

_pharmaceutics, 2020, doi:10.3390/pharmaceutics12100999_

Round 1
Reviewer 1 Report
The review focuses on formulation strategies towards mostly long-acting polypeptides via parenteral routes. However, the oral route is also discussed. This is an area that many good quality reviews are available and presented review offers limited novelty to existing body of work.
Please see my comments below:
- Abstract: Poor permeability across biological barriers is not stated as a challenge in peptide and protein therapeutics. I feel this needs to be exclusively stated.
- Introduction: “Proteins and peptides are fundamentally distinguished from each other by their differences in size.” Size refers to molecular volume, versus in the context of the text presented I think this mean molecular weight. Please amend.
- Clinically, non-endogenous peptides are more prevalent (e.g. glucagon-like peptide-1 (GLP-1) agonists, octreotide and goserelin) than non-endogenous proteins. This statement is misleading in my view. Stable analogues of endogenous peptides and proteins are clinically more prevalent. But as written it implies that non-endogenous i.e. synthetic peptides (which could be based on phage) are the norm. All of the examples are stable analogues and I think they need to be referred to as such, as these peptides still bind endogenous peptide receptor and mimic activity of endogenous peptide.
- Unclear as to what a “brain route” is at the last part of introduction.
- I feel that the aim of the review needs to be clearly described as oral delivery and non-parenteral routes are included but not all of the routes i.e. nasal or transdermal.
- “Peptides have the propensity to self-associate, forming fibrils and aggregates with consequently reduced activity and bioavailability”. This statement is not a rule of thumb as stated. Lanreotide is a clinical available peptide aggregate with improved bioavailability and clinical efficacy stemming from self-association.
- Section 2. Poor permeability across biological barriers is not clearly discussed in this section.
- Kidney glomeruli have a pore size of about 8 nm – Please reference
- Section 2. Challenges are presented under a general umbrella of “Pharmacokinetics” I think this section needs reorganization and being split into 4 subsections regarding absorption, distribution, metabolism, and elimination specifically per routes covered.
- “The mass of a therapeutic protein” is not equivalent to molecular weight of therapeutic protein but infers to the dose. Please amend
- Stategies to increase half-life do not include analogues (by modification of amino acids) or lipidisation. Please include them.
- Please summarise optimal PEG molecular weight needed to stabilise peptides and proteins depending on route of administration. Discuss toxicity, permeability and stabilization in terms of molecular weight of PEG chosen.
- Section 3.2.1 – Generalised summary not really extending from what is already widely known to audience of this journal. Providing an in depth table with novel formulations of peptides and proteins along with their particle characteristics and in depth w/w composition would be more beneficial.
- Section 3.2.2 – Same as point 13 above.
- Section 4. You refer to the routes and then you mention brain delivery. You could make a subsection in each route, but unless you are referring to direct administration in the brain this is not a different route of delivery. Please re-write.
- Table 1. Really offers very little as it not exhaustive and lacks info e.g. desmopressin oral tablets were amongst the first oral peptides but not mentioned; distribution of therapeutics in the brain is not mentioned as a challenge. Again “Brain delivery” is not a route of administration although is a target. But I cannot understand why a special case is made, when no real detail is given regarding optimal delivery of peptides and proteins to the brain apart from after ocular administration.
- Studies described in Section 4, should be summarized systematically in tables with full detail. This will shorten the review and make the information readily accessible to readers.
- Table 2: Instead of “effect” mechanism of release is more appropriate Also be specific in description regarding mechanism of release i.e. erosion from microparticles instead of general release from particulates. Also leuprolide can exist as monthly , 3 months and 6 months injection.
- “So far, only an oral formulation of semaglutide (Ozempic®), a GLP-1 receptor agonist for the treatment of type II diabetes, has been developed by Novo Nordisk.” This is not true - see DDAVP* Melt, cyclosporin. In this case you discuss lipidisation, however this strategy does not feature in the section above with different strategies.
- Oral section is poorly written and unclear. Figure 3 offers little if anything to the review. SEDDS should be included in this section as in Phase III.
- Oral to brain studies are not discussed.
- Section 4.4.2 about brain delivery - is unclear as to why is included. I think strategies for brain delivery of peptides and proteins should be presented in different routes. As presented is mixed up and really makes little sense. If the review is on peptide and protein delivery, specify routes and make a case about SC to brain, oral to brain, skin to brain etc. This section is generic and confuses order. Also nasal delivery is not discussed fully but appears only for nose to brain delivery although nasal delivery of peptide and proteins is an option even if the target is not the brain.
- I feel the review is extremely long, but only offers few novel points that have not been previously reviewed extensively and in more detail. I believe the authors should limit scope and tailor review to areas of strength, as currently is only presenting a selection of research (not clear how this fraction was selected) and is not an exhaustive systematic review to guide readers in the field. It is aimed to novices in the field but does not provide an up to date and in depth summary for researchers with background in the area. Depth is missing in many cases and reader is left to look at original work to obtain needed information.
Author Response
Reviewer 1
The review focuses on formulation strategies towards mostly long-acting polypeptides via parenteral routes. However, the oral route is also discussed. This is an area that many good quality reviews are available and presented review offers limited novelty to existing body of work.
Please see my comments below:
- Abstract: Poor permeability across biological barriers is not stated as a challenge in peptide and protein therapeutics. I feel this needs to be exclusively stated.
Response: The revised abstract is more concisely written while ensuring that poor permeability across biological barriers is mentioned as a challenge for polypeptide delivery.
- Introduction: “Proteins and peptides are fundamentally distinguished from each other by their differences in size.” Size refers to molecular volume, versus in the context of the text presented I think this mean molecular weight. Please amend.
Response: We have changed the word ‘size’ to ‘molecular volume’ in the revised introduction. The introduction was also edited.
- Clinically, non-endogenous peptides are more prevalent (e.g. glucagon-like peptide-1 (GLP-1) agonists, octreotide and goserelin) than non-endogenous proteins. This statement is misleading in my view. Stable analogues of endogenous peptides and proteins are clinically more prevalent. But as written it implies that non-endogenous i.e. synthetic peptides (which could be based on phage) are the norm. All of the examples are stable analogues and I think they need to be referred to as such, as these peptides still bind endogenous peptide receptor and mimic activity of endogenous peptide.
Response: The statement has been amended in the revised manuscript to reflect the Reviewer’s comment on the prevalence of non-endogenous peptides and proteins clinically.
- Unclear as to what a “brain route” is at the last part of introduction.
Response: We have amended the introduction in the revised manuscript, hence, ‘brain route’ is not present.
- I feel that the aim of the review needs to be clearly described as oral delivery and non-parenteral routes are included but not all of the routes i.e. nasal or transdermal.
Response: The aim of the review has been clearly clarified now in the revised introduction.
- “Peptides have the propensity to self-associate, forming fibrils and aggregates with consequently reduced activity and bioavailability”. This statement is not a rule of thumb as stated. Lanreotide is a clinical available peptide aggregate with improved bioavailability and clinical efficacy stemming from self-association.
Response: We have amended the text in the revised manuscript to include information about peptide self-assembly in drug delivery. The wording has also been modified to specify that numerous other peptides may aggregate and lose function when self-assembly occurs.
- Section 2. Poor permeability across biological barriers is not clearly discussed in this section.
Response: We agree and peptide and protein permeability is now discussed under the ‘General concepts and challenges’ subheading (Section 4) of the different routes of administration covered in the review instead of ‘General challenges in polypeptide delivery’ (Section 2).
- Kidney glomeruli have a pore size of about 8 nm – Please reference
Response: We have added a new reference in the revised manuscript to support the above statement:
Longmire M, Choyke PL, Kobayashi H. Clearance properties of nano-sized particles and molecules as imaging agents: considerations and caveats. Nanomedicine. 2008;3(5):703-717. doi:10.2217/17435889.3.5.703
- Section 2. Challenges are presented under a general umbrella of “Pharmacokinetics” I think this section needs reorganization and being split into 4 subsections regarding absorption, distribution, metabolism, and elimination specifically per routes covered.
Response: We have edited and reorganised Section 2 to ensure there is good balance with the bulk of the review, which is focused on the specific challenges extend drug action for the common routes of administration and to provide an overview of the state of the art in implantable technologies. We have described the key pharmacokinetic issues citing the more detailed and specialised reviews focused on pharmacokinetics.
- “The mass of a therapeutic protein” is not equivalent to molecular weight of therapeutic protein but infers to the dose. Please amend
Response: We have changed the word ‘mass’ to ‘molecular weight’ in the text as suggested by the Reviewer.
- Strategies to increase half-life do not include analogues (by modification of amino acids) or lipidisation. Please include them.
Response: We fully agree and were remiss in our submitted manuscript. Amino acid modification is now included in the last part of paragraph of Section 3.1 (lines 1082-1084 in the marked-up revised manuscript). Lipidisation is clarified in the first paragraph on page 7 pf the revised manuscript.
- Please summarise optimal PEG molecular weight needed to stabilise peptides and proteins depending on route of administration. Discuss toxicity, permeability and stabilization in terms of molecular weight of PEG chosen.
Response: Protein PEGylation has been thoroughly described in detail in many reviews and we cite recent comprehensive sources. We have revised the text related to PEGylation to ensure the key concepts, and advantages and disadvantages are described concisely. Considerable detailed discussion of protein PEGylation is required to differentiate differences in PEG molecular weights used, e.g. ~ 5 kDa for hyperPEGylated non-human proteins and anywhere from 12-60 kDa molar amounts of PEG for mono-PEGylated conjugates from a wide range of proteins including from cytokines, hematopoietic agents, blood factors, hormones and Fabs. We are concerned such a discussion would take this review out of scope and since the review is long already, our efforts were focused to edit the review while ensuring the review addressed the specific current challenges and strategies to develop long acting polypeptide forms.
- Section 3.2.1 – Generalised summary not really extending from what is already widely known to audience of this journal. Providing an in depth table with novel formulations of peptides and proteins along with their particle characteristics and in depth w/w composition would be more beneficial.
Response: We have extensively edited Section 3.2.1 as we were not trying to give a detailed review of particulate associated formulations or the fabrication processes used as the field is so large with a considerable number of recent reviews having been written. Our intent was to provide a brief description of the main underlying concepts to help with our focus on routes of administration where we go into more considered detail
- Section 3.2.2 – Same as point 13 above.
Response: We have also extensively edited Section 3.2.2 with the same goals in mind as we edited Section 3.2.1. Since there are recent detailed reviews about gels, we did not want to give the reader the sense that we were trying to provide yet another review about gels. Section 3 was written to provide the review the balance and definition of terms needed to go into the detail of Section 4.
- Section 4. You refer to the routes and then you mention brain delivery. You could make a subsection in each route, but unless you are referring to direct administration in the brain this is not a different route of delivery. Please re-write.
Response: We understand and in the revised manuscript, Section 4 is titled Polypeptide delivery. It is now split into two subsections: 4.1 Routes of administration- that covers intravenous, subcutaneous, oral and mucosal, and ocular drug delivery; and 4.l.2 Targets-that covers targeting the brain.
- Table 1. Really offers very little as it not exhaustive and lacks info e.g. desmopressin oral tablets were amongst the first oral peptides but not mentioned; distribution of therapeutics in the brain is not mentioned as a challenge. Again “Brain delivery” is not a route of administration although is a target. But I cannot understand why a special case is made, when no real detail is given regarding optimal delivery of peptides and proteins to the brain apart from after ocular administration.
Response: We have updated Table 1 and we have also included desmopressin. Each subsection in Section 4 is more exhaustive with a significant amount of detail, examples and references. In the revised version of Table 1, we have highlighted subcutaneous, oral and ocular under routes of delivery; and targeting the brain under targets of delivery. We wanted to mention the brain as a target in our review as significant research has been conducted over the years for delivering polypeptides to that target.
- Studies described in Section 4, should be summarized systematically in tables with full detail. This will shorten the review and make the information readily accessible to readers.
Response: We have tried our best as much information as possible to be more accessible in the revised manuscript, for example, we have added a new table (Table 3) in Section 4.1.3-oral and mucosal drug delivery.
- Table 2: Instead of “effect” mechanism of release is more appropriate Also be specific in description regarding mechanism of release i.e. erosion from microparticles instead of general release from particulates. Also leuprolide can exist as monthly , 3 months and 6 months injection.
Response: We have amended the table heading as ‘release mechanism’ instead of ‘effect’. We have also added other possible dosing durations for leuprolide.
- “So far, only an oral formulation of semaglutide (Ozempic®), a GLP-1 receptor agonist for the treatment of type II diabetes, has been developed by Novo Nordisk.” This is not true - see DDAVP* Melt, cyclosporin. In this case you discuss lipidisation, however this strategy does not feature in the section above with different strategies.
Response: We have modified the statement to specify that semaglutide is the only long-acting oral formulation available for type II diabetes.
- Oral section is poorly written and unclear. Figure 3 offers little if anything to the review. SEDDS should be included in this section as in Phase III.
Response: We have improved the oral section in the revised manuscript and have also introduced Table 3 to make some of the information more accessible. We have also included SEDDs under this section.
- Oral to brain studies are not discussed.
Response: In the revised manuscript, we have made a separate subheading called “Targets” (Section 4.2) and “Targeting the brain” is under Section 4.2.1. Within that section, general strategies and key examples of polypeptide delivery to the brain are discussed. Other routes such as nasal delivery of polypeptides to the brain are also highlighted.
- Section 4.4.2 about brain delivery - is unclear as to why is included. I think strategies for brain delivery of peptides and proteins should be presented in different routes. As presented is mixed up and really makes little sense. If the review is on peptide and protein delivery, specify routes and make a case about SC to brain, oral to brain, skin to brain etc. This section is generic and confuses order. Also nasal delivery is not discussed fully but appears only for nose to brain delivery although nasal delivery of peptide and proteins is an option even if the target is not the brain.
Response: We thank the Reviewer for this comment and in the revised manuscript, the brain as a target is a separate subsection (4.2.1) to avoid reader confusion.
- I feel the review is extremely long, but only offers few novel points that have not been previously reviewed extensively and in more detail. I believe the authors should limit scope and tailor review to areas of strength, as currently is only presenting a selection of research (not clear how this fraction was selected) and is not an exhaustive systematic review to guide readers in the field. It is aimed to novices in the field but does not provide an up to date and in depth summary for researchers with background in the area. Depth is missing in many cases and reader is left to look at original work to obtain needed information.
Response: We have extensively edited the revised manuscript to shorten the review and make the information more accessible for the readers. We have addressed all of the Reviewer’s comments, and we hope the quality of the revised manuscript is improved.

Reviewer 2 Report
Polypeptides including proteins and peptides have been widely used to treat various diseases such as cancer, bacterial infections, inflammation, and autoimmune disease. However, the disadvantages of polypeptides include short half-lives, poor pharmacokinetics, poor pharmacodynamics, and risk for immunogenicity. This manuscript represents a comprehensive review for polypeptide-based drug delivery and discusses the challenges in therapeutic applications. In addition, the authors provide formulation strategies for different routes such as intravenous (IV), subcutaneous (SC), oral, brain, and ocular routes. Overall, this manuscript is well organized and integrates different strategies for the development of drug delivery and formulation. It is suitable for published on Pharmaceutics.
Minor comments
- There are some abbreviations that are not explained at their first appearance, such as APIs, PEG, etc.
Author Response
Polypeptides including proteins and peptides have been widely used to treat various diseases such as cancer, bacterial infections, inflammation, and autoimmune disease. However, the disadvantages of polypeptides include short half-lives, poor pharmacokinetics, poor pharmacodynamics, and risk for immunogenicity. This manuscript represents a comprehensive review for polypeptide-based drug delivery and discusses the challenges in therapeutic applications. In addition, the authors provide formulation strategies for different routes such as intravenous (IV), subcutaneous (SC), oral, brain, and ocular routes. Overall, this manuscript is well organized and integrates different strategies for the development of drug delivery and formulation. It is suitable for published on Pharmaceutics.
Minor comments
- There are some abbreviations that are not explained at their first appearance, such as APIs, PEG, etc.
Response: We have now defined all abbreviations in the revised manuscript.
Reviewer 3 Report
Overall, the manuscript is well-written and comprehensive. The authors took great care of writing the manuscript so that the manuscript is free of grammatical or typographical errors. The reviewer would like to make a couple of minor suggestions for the broad readership of Pharmaceutics.
Suggestion #1: Since the article is extensive in its contents and lengthy, it will help the readers greatly if the authors include the list of contents at the beginning of the manuscript, such as the one shown below.
- Introduction
- General challenges in polypeptide delivery
2.1 Aggregation
2.2 Pharmacokinetics
- General strategies to increase duration of action
3.1 Half-life extension strategies
3.1.1 Recycling
Fc-fusions
3.1.2 Increasing size and shielding effects
Glycoengineering
Conjugation to PEG
Beyond PEGylation
Conjugation to albumin
3.1.3 Targeting tissue components
Affinity-based drug delivery
3.2 Depot formulation strategies to prolong duration of action
3.2.1 Particulate formulations
3.2.2 Gels
- Routes of drug delivery
4.1 IV drug delivery
4.1.1 General issues and challenges
4.1.2 IV formulation strategies
4.2 SC drug delivery
4.2.1 General concepts and challenges
4.2.2 SC formulation strategies
4.3 Oral and mucosal drug delivery
4.3.1 General concepts and challenges
4.3.2 Oral and mucosal-formulation strategies
4.4 Brain delivery
4.4.1. General concepts and challenges
4.4.2 Brain formulation strategies
4.5 Ocular drug delivery
4.5.1 General concepts and challenges
4.5.2 Ocular formulation strategies
- Conclusions
-----------------------------
Suggestion #2:
Please undo italicization of the section title ‘3.2 Depot formulation strategies to prolong duration of action’. All other headings are not italicized.
Suggestion #3:
Instead of placing the section ‘3.1.3 Targeting tissue components’ under one of the strategies to extending half-lives per se, this could be treated separately under Section 3 (for example, targeting could be section 3.3 after the Depot formulations strategies). In the reviewer’s humble opinion, this revision makes better sense.
Suggestion #4:
Under the section 3.1.2, there is a subsection entitled ‘Conjugation to albumin’. However, the content covers both noncovalent and covalent attachment strategies (including HSA fusion protein approaches). The reviewer suggests the revision of the subheading to, for example, ‘Attachment to albumin’.
Suggestion #5:
Although the manuscript covers a substantial number of examples on extending the half-lives polypeptide therapeutics, RBC-mediated delivery is probably another emerging area to this end, which is not covered. Instead of writing a section about the subtopic, since the paper is already lengthy, the authors could simply cite and direct the readers to a review on RBC carriers as a long-circulating DDS toward the end of the section 3.1.2.
Author Response
Reviewer 3
Overall, the manuscript is well-written and comprehensive. The authors took great care of writing the manuscript so that the manuscript is free of grammatical or typographical errors. The reviewer would like to make a couple of minor suggestions for the broad readership of Pharmaceutics.
- Suggestion #1: Since the article is extensive in its contents and lengthy, it will help the readers greatly if the authors include the list of contents at the beginning of the manuscript, such as the one shown below.
Introduction
General challenges in polypeptide delivery
2.1 Aggregation
2.2 Pharmacokinetics
General strategies to increase duration of action
3.1 Half-life extension strategies
3.1.1 Recycling
Fc-fusions
3.1.2 Increasing size and shielding effects
Glycoengineering
Conjugation to PEG
Beyond PEGylation
Conjugation to albumin
3.1.3 Targeting tissue components
Affinity-based drug delivery
3.2 Depot formulation strategies to prolong duration of action
3.2.1 Particulate formulations
3.2.2 Gels
Routes of drug delivery
4.1 IV drug delivery
4.1.1 General issues and challenges
4.1.2 IV formulation strategies
4.2 SC drug delivery
4.2.1 General concepts and challenges
4.2.2 SC formulation strategies
4.3 Oral and mucosal drug delivery
4.3.1 General concepts and challenges
4.3.2 Oral and mucosal-formulation strategies
4.4 Brain delivery
4.4.1. General concepts and challenges
4.4.2 Brain formulation strategies
4.5 Ocular drug delivery
4.5.1 General concepts and challenges
4.5.2 Ocular formulation strategies
Conclusions
Response: We thank the Reviewer for their comment. We have included a Table of Contents in the revised manuscript in accordance with the Pharmaceutics MDPI Instructions for Authors.
- Suggestion #2: Please undo italicization of the section title ‘3.2 Depot formulation strategies to prolong duration of action’. All other headings are not italicized.
Response: We have followed the template provided by Pharmaceutics MDPI journal and the italicisation is from the font embedded within the template.
- Suggestion #3: Instead of placing the section ‘3.1.3 Targeting tissue components’ under one of the strategies to extending half-lives per se, this could be treated separately under Section 3 (for example, targeting could be section 3.3 after the Depot formulations strategies). In the reviewer’s humble opinion, this revision makes better sense.
Response: We have made the above revision requested by the Reviewer and shifted ‘Targeting tissue components’ to a new section (Section 3.3).
- Suggestion #4: Under the section 3.1.2, there is a subsection entitled ‘Conjugation to albumin’. However, the content covers both noncovalent and covalent attachment strategies (including HSA fusion protein approaches). The reviewer suggests the revision of the subheading to, for example, ‘Attachment to albumin’.
Response: We have revised the subheading as ‘Polypeptide conjugation to a water soluble macromolecule’, which covers PEGylation and non-covalent and covalent attachment strategies.
- Suggestion #5: Although the manuscript covers a substantial number of examples on extending the half-lives polypeptide therapeutics, RBC-mediated delivery is probably another emerging area to this end, which is not covered. Instead of writing a section about the subtopic, since the paper is already lengthy, the authors could simply cite and direct the readers to a review on RBC carriers as a long-circulating DDS toward the end of the section 3.1.2.
Response: We have introduced RBC conjugation at the end of Section 3.1.2 and added the following new references:
- Muzykantov VR. Drug delivery by red blood cells: vascular carriers designed by mother nature. Expert Opin Drug Deliv. 2010;7(4):403-427. doi:10.1517/17425241003610633
- Rossi L, Fraternale A, Bianchi M, Magnani M. Red Blood Cell Membrane Processing for Biomedical Applications. Front Physiol. 2019;10(August):1-8. doi:10.3389/fphys.2019.01070
- Kontos S, Kourtis IC, Dane KY, Hubbell JA. Engineering antigens for in situ erythrocyte binding induces T-cell deletion. Proc Natl Acad Sci U S A. 2013;110(1). doi:10.1073/pnas.1216353110
- Villa CH, Pan DC, Zaitsev S, Cines DB, Siegel DL, Muzykantov VR. Delivery of drugs bound to erythrocytes: new avenues for an old intravascular carrier. Ther Deliv. 2015;6(7):795-826. doi:10.4155/tde.15.34
- Glassman PM, Villa CH, Ukidve A, et al. Vascular drug delivery using carrier red blood cells: Focus on RBC surface loading and pharmacokinetics. Pharmaceutics. 2020;12(5):1-21. doi:10.3390/pharmaceutics12050440
- Xia Q, Zhang Y, Li Z, Hou X, Feng N. Red blood cell membrane-camouflaged nanoparticles: a novel drug delivery system for antitumor application. Acta Pharm Sin B. 2019;9(4):675-689. doi:10.1016/j.apsb.2019.01.011
